# QUANTIZED VISUAL GEOMETRY GROUNDED TRANSFORMER

**Weilun Feng**[1,2]*, **Haotong Qin**[3]*, **Mingqiang Wu**[1,2]*, **Chuanguang Yang**[1]†, **Yuqi Li**[4],
**Xiangqi Li**[1,2], **Zhulin An**[1]†, **Libo Huang**[1], **Yulun Zhang**[5], **Michele Magno**[3], **Yongjun Xu**[1]

[1]State Key Laboratory of AI Safety, Institute of Computing Technology, Chinese Academy of Sciences
[2]University of Chinese Academy of Sciences   [3]ETH Zürich
[4]City College of New York, City University of New York, USA   [5]Shanghai Jiao Tong University
{fengweilun24s,yangchuanguang,lixiangqi24s,anzhulin,xyj}@ict.ac.cn
{haotong.qin,michele.magno}@pbl.ee.ethz.ch, wumingqiang25@mails.ucas.ac.cn,
{yuqili010602,www.huanglibo,yulun100}@gmail.com

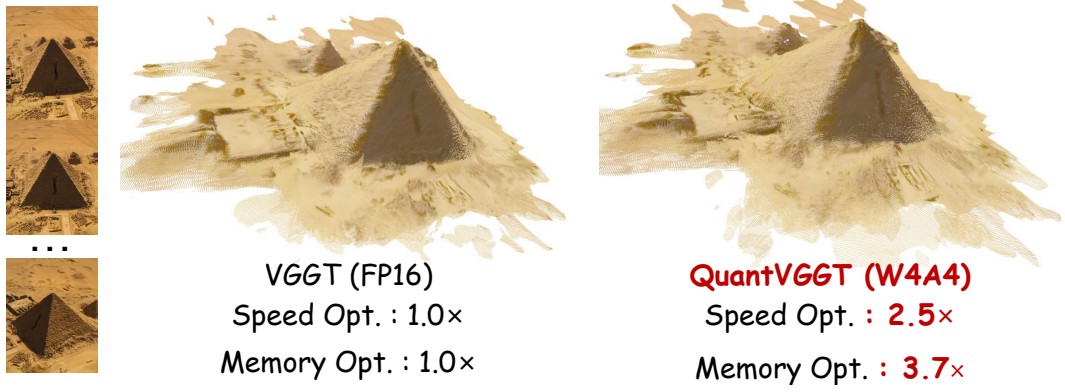

VGGT (FP16)
Speed Opt. : 1.0×
Memory Opt. : 1.0×

**QuantVGGT (W4A4)**
Speed Opt. : **2.5×**
Memory Opt. : **3.7×**

Figure 1: **QuantVGGT** effectively quantizes VGGT (Wang et al., 2025a) to W4A4 without compromising visual quality while bringing **2.5×** speedup and **3.7×** compression.

## ABSTRACT

Learning-based 3D reconstruction models, represented by Visual Geometry Grounded Transformers (VGGTs), have made remarkable progress with the use of large-scale transformers. Their prohibitive computational and memory costs severely hinder real-world deployment. Post-Training Quantization (PTQ) has become a common practice for compressing and accelerating models. However, we empirically observe that PTQ faces unique obstacles when compressing billion-scale VGGTs: the data-independent special tokens induce heavy-tailed activation distributions, while the multi-view nature of 3D data makes calibration sample selection highly unstable. This paper proposes the first **Quant**ization framework for **VGGT**s, namely *QuantVGGT*. This mainly relies on two technical contributions: First, we introduce *Dual-Smoothed Fine-Grained Quantization*, which integrates pre-global Hadamard rotation and post-local channel smoothing to mitigate heavy-tailed distributions and inter-channel variance robustly. Second, we design *Noise-Filtered Diverse Sampling*, which filters outliers via deep-layer statistics and constructs frame-aware diverse calibration clusters to ensure stable quantization ranges. Comprehensive experiments demonstrate that QuantVGGT achieves the state-of-the-art results across different benchmarks and bit-width, surpassing the previous state-of-the-art generic quantization method with a great margin. We highlight that our 4-bit QuantVGGT can deliver a **3.7×** memory reduction and **2.5×** acceleration in real-hardware inference,

---

*Equal contribution.
†Corresponding authors: Zhulin An, anzhulin@ict.ac.cn; Chuanguang Yang, yangchuanguang@ict.ac.cn

while maintaining reconstruction accuracy above **98%** of its full-precision counterpart. This demonstrates the vast advantages and practicality of QuantVGGT in resource-constrained scenarios. Our code is released in https://github.com/wlfeng0509/QuantVGGT.

# 1 INTRODUCTION

Recent advances in learning-based 3D reconstruction have demonstrated unprecedented capabilities in recovering dense geometry and camera trajectories directly from image sequences. Traditional approaches (Mur-Artal et al., 2015; Mur-Artal & Tardós, 2017; Schonberger & Frahm, 2016; Hartley & Zisserman, 2003) are grounded in geometric priors and optimization, but their reliance on hand-crafted design choices and iterative solvers often leads to limited scalability and reduced robustness in complex scenes. In contrast, large-scale deep models have shifted the paradigm toward data-driven frameworks, offering remarkable generalization ability across diverse environments (Wang et al., 2025b; Yang et al., 2025). A milestone in this evolution is the Visual Geometry Grounded Transformer (VGGT) (Wang et al., 2025a). This 1.2B-parameter model unifies multiple 3D tasks, including dense depth estimation, point map regression, camera pose prediction, and point tracking within a single forward pass, consistently surpassing task-specialized counterparts.

Despite its success, the billion-scale parameterization of VGGT incurs prohibitive computational and memory costs, severely restricting its deployment in real-world scenarios. Model quantization (Gholami et al., 2022; Jacob et al., 2018) is an effective compression technique by converting weights and activations of model from high-precision floating-points to low-precision integers. While this technique has been widely validated in large language models (Frantar et al., 2022; Xiao et al., 2023) and 2D vision models (Yuan et al., 2022; Wu et al., 2024), the quantization of billion-scale 3D reconstruction transformers such as VGGT remains largely unexplored. In our study, we identify two model-specific properties of VGGT that make its quantization particularly challenging: ❶ **The presence of data-independent special tokens (camera and register tokens).** Unlike regular image tokens that are encoded from input images, these tokens are pretrained and injected into image tokens to encode global context and cross-view geometry. This data-independent property causes activation distributions to deviate from typical patterns, amplifying heavy tails and producing extreme channel and token variance. These skewed statistics are unfriendly to standard quantization, leading to substantial information loss. ❷ **The inherently semantic complexity of 3D data.** Each input sequence involves non-identical and complex views, meaning that the underlying semantic space is both high-dimensional and highly redundant. For quantization calibration, the ideal process is to perceive the expected major data distribution. If calibration samples are rare outliers and not diverse, the estimated quantization ranges become biased and fail to generalize, causing performance degradation across unseen scenes. Thus, sample diversity and representativeness are far more critical than in 2D vision tasks.

To address these challenges, we present the first systematic investigation of Post-Training Quantization (PTQ) for VGGT and propose a tailored framework, **QuantVGGT**. Our approach introduces *Dual-Smoothed Fine-Grained Quantization (DSFQ)*, which mitigates skewed statistics by combining (1) a *pre-global rotation* via Hadamard transforms to disperse outliers and smooth heavy-tailed distributions, and (2) a *post-local smoothing* step that normalizes channel-level variance in the rotated space. Additionally, to overcome calibration instability, we design *Noise-Filtered Diverse Sampling (NFDS)*, which leverages deep-layer activation statistics to filter noisy extremes and employs frame-aware clustering aligned with VGGT's inductive biases. Together, these components yield robust, efficient, and accurate quantization of billion-scale 3D reconstruction transformers.

Our contributions are summarized as follows:

1. We provide the first systematic analysis of PTQ on VGGT, highlighting quantization challenges rooted in its data-independent tokens and multi-view activation statistics.

2. We propose a dual-stage smoothing scheme that globally disperses heavy-tailed distributions and locally balances channel variance, significantly reducing quantization errors.

3. We design a calibration strategy that filters outliers and utilizes VGGT's inductive bias to construct frame-aware clusters, ensuring a representative and stable calibration set.

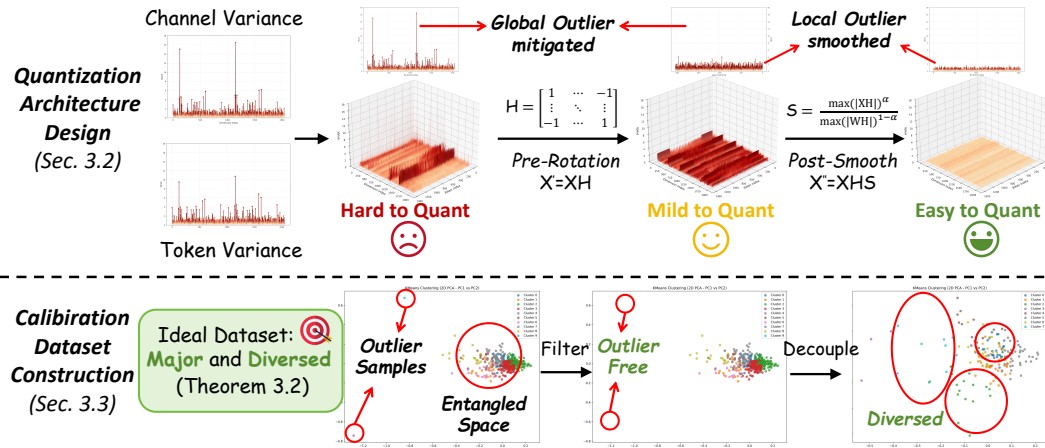

Figure 2: **Overview of proposed QuantVGGT. Top**: Our proposed Dual-Smoothed Fine-Grained Quantization architecture. **Bottom**: Our proposed Noise-Filtered Diverse Sampling strategy.

4. Extensive experiments demonstrate that our approach enables effective low-bit quantization of VGGT, achieving substantial memory and inference efficiency gains while preserving reconstruction accuracy.

## 2 RELATED WORKS

### 2.1 LEARNING-BASED 3D RECONSTRUCTION

Thanks to the development of deep learning technology in recent years, 3D reconstruction tasks have gradually shifted from traditional methods (Mur-Artal & Tardós, 2017; Mur-Artal et al., 2015; Schonberger & Frahm, 2016; Hartley & Zisserman, 2003; An et al., 2025; Li et al., 2026) that rely heavily on prior knowledge to data-driven learning-based methods. Due to the large-scale training process, learning-based methods (Wang et al., 2025b; Yang et al., 2025) often achieve better reconstruction performance and generalization ability. DUSt3R (Wang et al., 2024) predicts the 3D point maps of a scene by regressing two RGB images, laying the foundation for learning-based methods. MASt3R (Leroy et al., 2024) further refines the framework by introducing confidence-weighted losses for metric scale approximation. Current model, VGGT (Wang et al., 2025a) enables predicting camera position, dense depth, point maps, and point tracking with a single forward process. With scaling-up to 1.2B parameters, VGGT achieves state-of-the-art results across various 3D tasks with even surpasses some task-specified methods. But up to billions of parameters and enormous computational complexity of VGGT severely limit its widespread deployment and application. However, the compression methods for VGGT, such as quantization, are still highly unexplored.

### 2.2 MODEL QUANTIZATION

Model quantization (Gholami et al., 2022; Krishnamoorthi, 1806; Ma et al., 2024a; 2023; 2024b; Feng et al., 2025b) significantly reduces the memory footprint and accelerates inference by reducing the data bit-width. Model quantization can be mainly divided into Quantization-Aware Training (QAT) and Post-Training Quantization (PTQ). QAT (Jacob et al., 2018; Qin et al., 2020b; Feng et al., 2025a;d) utilizes substantial data to train quantization parameters and model weights, thus typically ensuring good performance at extremely low bits. But QAT often requires massive training resources. On the contrary, PTQ (Wei et al., 2024; Frantar et al., 2022) only requires little calibration data to fine-tune the quantization parameters, and therefore can be applied to large models. For PTQ, BRECQ (Li et al., 2021) builds the block-wise reconstruction framework, and QDrop (Wei et al., 2022) further enhances the performance by randomly dropping quantization activations. To ensure the effectiveness of PTQ on large models, GPTQ (Frantar et al., 2022) utilizes approximate second-order gradient to optimize Large Language Models. To address the impact of imbalanced distribution on quantization, SmoothQuant (Xiao et al., 2023) introduces a smoothing parameter to transfer the difficulty of activation quantization to weight. QuaRot (Ashkboos et al., 2024) adopts

a similar rotation to smooth the distribution. Although these methods perform well on existing 2D-visual and language models, they do not generalize well to large-scale 3D models like VGGT (Wang et al., 2025a). To the best of our knowledge, our method is the first PTQ framework specially designed for VGGT, ensuring its performance even at low-bit quantization.

## 3 METHODS

### 3.1 PRELIMINARY

#### 3.1.1 VISUAL GEOMETRY GROUNDED TRANSFORMER

Visual Geometry Grounded Transformer (VGGT) (Wang et al., 2025a) is a recent architecture designed to predict all key 3D attributes from image sequences of arbitrary length. Its core components are *tokenization* and *token registration*. For any input sequence $\mathcal{I} = \{I_i\}_{i=1}^{N}$ of $N$ RGB frames, VGGT first tokenizes each frame using a pretrained vision backbone $\mathcal{F}(\cdot)$, such as DINOv2 (Oquab et al., 2023), producing

$$\mathcal{X} = \{x_i \mid x_i = \mathcal{F}(I_i)\}_{i=1}^{N}, \quad x_i \in \mathbb{R}^{n \times d}, \tag{1}$$

where $n$ denotes the token length after patching and $d$ is the feature dimension.

To enable multi-attribute reasoning, VGGT augments each frame with one *camera token* and four *register tokens*, which are responsible for aggregating different 3D attributes (e.g., camera parameters, scene geometry). Notably, VGGT introduces two distinct sets of these special tokens: one set $t_{\mathrm{f}} \in \mathbb{R}^{5 \times d}$ is reserved for the first frame, while another set $t_{\mathrm{o}} \in \mathbb{R}^{5 \times d}$ is shared by all subsequent frames. Formally, the token registration process is defined as

$$\hat{\mathcal{X}} = \{\hat{x}_i \mid \hat{x}_1 = \mathrm{concat}(x_1, t_{\mathrm{f}}), \ \hat{x}_{k \neq 1} = \mathrm{concat}(x_k, t_{\mathrm{o}})\}_{i=1}^{N}, \tag{2}$$

and the resulting $\hat{\mathcal{X}}$ is then forwarded into the VGGT backbone.

#### 3.1.2 POST-TRAINING QUANTIZATION

Quantization (Gholami et al., 2022; Krishnamoorthi, 1806) aims to convert model weights and activations from floating-point representations into compact low-bit integer formats, thereby reducing both computational cost and memory footprint. Formally, given a floating-point vector $x$, the symmetric quantization procedure can be described as:

$$x_{\mathrm{int}} = \mathrm{clamp}\left(\mathrm{round}\left[\tfrac{x}{\Delta}\right], -2^{N-1}, 2^{N-1} - 1\right), \quad \Delta = \tfrac{\max(|x|)}{2^{N-1}-1}, \tag{3}$$

where $N$ represents the target bit-width, $\mathrm{round}(\cdot)$ denotes the rounding operator, and $\mathrm{clamp}(\cdot)$ ensures that the integer values remain within the valid range $[-2^{N-1}, 2^{N-1} - 1]$.

Among quantization paradigms, Post-Training Quantization (PTQ) (Wei et al., 2024; Frantar et al., 2022; Feng et al., 2025c) is widely applied for its efficiency. Unlike Quantization-Aware Training (Qin et al., 2020a; Feng et al., 2025a;d), PTQ does not require fine-tuning the weights. Instead, it fine-tunes the quantization parameters using only a relatively small calibration dataset $\mathcal{D}_{\mathrm{calib}}$, while keeping the original full-precision weights fixed. This makes PTQ particularly attractive in real-world deployment where computational resources for fine-tuning are limited.

Following the standard practice in prior works (Yuan et al., 2022; Shang et al., 2023; Feng et al., 2024; Xiao et al., 2023), the quantization error is typically measured by the following objective:

$$\mathcal{L}_{\mathrm{quant}} = \mathbb{E}_{x \sim \mathcal{D}_{\mathrm{calib}}}\left[||\theta^f(x) - \theta^q(x)||_2^2\right], \tag{4}$$

where $\theta^f$ and $\theta^q$ denote the full-precision and quantized model functions, respectively.

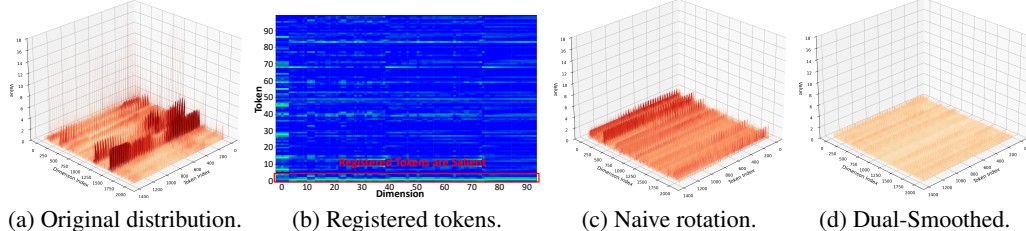

(a) Original distribution.     (b) Registered tokens.     (c) Naive rotation.     (d) Dual-Smoothed.

Figure 3: **The motivation and effect of Dual-Smoothed Fine-Grained Quantization.** (a): Salient distribution of VGGT (Wang et al., 2025a) *frame_block 9.* (b):Saliency of registered tokens. The first five special tokens are more salient. (c): Distribution after naive rotation. (d): Distribution after our dual-smooth. **We provide more analysis in Appendix Sec. D.**

## 3.2 DUAL-SMOOTHED FINE-GRAINED QUANTIZATION

**Observation 1.** *VGGT (Wang et al., 2025a) exhibits highly skewed numerical distributions, which are amplified by data-independent tokens (camera and register tokens), leading to substantial quantization errors.*

As illustrated in Fig. 3b, these data-independent tokens (first 5 tokens) amplify channel and token numerical variance: with massive outliers that are much larger than regular patch tokens, producing heavy-tailed distributions. When passed into quantization, these few large elements occupy most of the quantization bins, causing severe numerical distortion (Xiao et al., 2023; Ashkboos et al., 2024).

**Pre-Global-Rotation.** Motivated by rotation-based quantization (Ashkboos et al., 2024; Zhao et al., 2024), we apply a random Hadamard transformation to spread out the impact of special-token-induced outliers. Given a target dimension $d = 2^n$, we can generate the corresponding Hadamard matrix $\hat{\mathbf{H}}_{2^n}$ through

$$\hat{\mathbf{H}}_2 = \frac{1}{\sqrt{2}} \begin{bmatrix} 1 & 1 \\ 1 & -1 \end{bmatrix} \text{ and } \hat{\mathbf{H}}_{2^n} = \hat{\mathbf{H}}_2 \otimes \hat{\mathbf{H}}_{2^{n-1}}. \tag{5}$$

For dimension that $d \neq 2^n$, we can factorize $d = 2^n m$ and have $\hat{\mathbf{H}}_d = \hat{\mathbf{H}}_{2^n} \otimes \hat{\mathbf{H}}_m$. Following (Ashkboos et al., 2024; Chee et al., 2023), we denote $v$ as a vector containing random draws from $\{+1, -1\}$, and a random Hadamard matrix $\mathbf{H} = \hat{\mathbf{H}}\text{diag}(v)$. This random Hadamard matrix satisfies $\mathbf{H}^\top \mathbf{H} = \mathbf{I}$. Given activation $\mathbf{X} \in \mathbb{R}^{n \times d_{in}}$ and weight $\mathbf{W} \in \mathbb{R}^{d_{out} \times d_{in}}$, the matrix multiplication invariance is preserved as follows:

$$\mathbf{X}\mathbf{W}^\top = (\mathbf{X}\mathbf{H})(\mathbf{W}\mathbf{H})^\top. \tag{6}$$

**Lemma 3.1.** *Due to the central limit effect, the distribution of values after Hadamard rotation tends to approximate a Gaussian, thereby smoothing the heavy-tailed distribution introduced by special tokens (Tseng et al., 2024).*

Lemma 3.1 suggests that the Hadamard rotation disperses outlier values across channels, resulting in a more uniform distribution, thereby significantly reducing their impact. Therefore, the original distribution becomes concentrated and smoother, which is more favorable for quantization. Figure 3c illustrates the smoothed distributions after the Hadamard rotation, where the extremely massive outliers are mitigated.

**Post-Local-Smooth.** Although the Hadamard rotation mitigates global skew, the transformed distribution still exhibits considerable local variance, as shown in Fig. 3c. While the Hadamard rotation spreads outliers across channels, it only weakens the global outliers, rather than eliminating the outliers within individual channels. To further reduce quantization error, we introduce a channel-wise scale to normalize the internal channel distributions. Unlike traditional scaling (Xiao et al., 2023; Wu et al., 2024) that computes $c_i = \frac{\max(|\mathbf{X}_i|)^\alpha}{\max(|\mathbf{W}_i|)^{1-\alpha}}$, we derive scale factors from the rotated distribution, ensuring robustness against extreme special-token values.

$$\hat{c}_i = \frac{\max(|\mathbf{X}_i\mathbf{H}|)^\alpha}{\max(|\mathbf{W}_i\mathbf{H}|)^{1-\alpha}}, \quad \mathbf{X}\mathbf{W}^\top = (\mathbf{X}\mathbf{H}\text{diag}(\hat{c})^{-1})(\text{diag}(\hat{c})\mathbf{H}^\top\mathbf{W}^\top), \tag{7}$$

where $\alpha$ balances quantization difficulty between activations and weights (typically $\alpha = 0.5$). This design offers two advantages: (1) the scale factor is derived from a smoother distribution after the pre-rotation, avoiding extreme values that could otherwise complicate weight quantization; and (2) it ensures that the post-scaled distribution even smoother. If using pre-scale, the post-rotation would destabilize the benefits of channel-scaling. The scale factors can also be fused into neighboring normalization layers (Xiao et al., 2023), introducing no runtime cost.

**Fine-Grained Quantization Granularity.** The above rotate-and-scale quantization strategy reduces quantization error by addressing the inner-dimension $d_{in}$. However, the choice of quantization granularity also plays a critical role in determining the overall error. Recent studies (Chee et al., 2023; Tseng et al., 2024) define the quantization difficulty using the concept of '$\mu$-coherent', where for any $x$, if $\max(x) \leq \mu\|x\|_F/\sqrt{g}$, with $g$ representing the number of elements, where $\mu$ represents the quantization difficulty. This suggests that reducing quantization granularity, when feasible, can significantly lower quantization error. From a hardware perspective, as long as the quantized matrix multiplication shares the same quantization parameters across the summation operation, there is no need to convert integers back to floating-point numbers, ensuring efficiency. In matrix multiplication, only the inner-channel $d_{in}$ values are summed. Therefore, we can utilize the outer-dimension $d_{out}$ for weight quantization and the token dimension $n$ for activation quantization. In practice, we apply out-dimension-wise quantization to the weights and token-wise quantization to the activations.

As shown in Fig. 3d, the proposed dual-smoothed fine-grained quantization further reduces the outer-dimension variance in the data distribution, significantly lowering the quantization error, with nearly no additional computational burden **(see Appendix Sec. D for efficiency analysis)**.

### 3.3 NOISE-FILTERED DIVERSE SAMPLING

The purpose of the PTQ calibration process is to approximate the behavior of the model in the real data distribution $\mathcal{X}$ using a small calibration set $\mathcal{D}_{\text{calib}}$. Formally, we seek

$$\theta_q^* = \arg\min_{\theta_q} \mathbb{E}_{x\sim\mathcal{X}}\left[\|\theta_f(x) - \theta_q(x)\|_2^2\right], \qquad (8)$$

and in practice we approximate the outer expectation with samples from $\mathcal{D}_{\text{calib}}$. Therefore, the calibration set should be statistically representative of $\mathcal{X}$.

**Theorem 3.2** (Calibration sampling principle). *Suppose $\mathcal{X}$ can be divided into different domains $\mathcal{X} = \{X_0, X_1, \cdots\}$. Each sub-domain $X_i$ has scale $V^i$ and can be partitioned into $N^i(\geq 2$ and finite) disjoint sub-regions denoted as $\{R_1^i, \cdots, R_{N^i}^i\}$ with corresponding scales $\{V_1^i, \cdots, V_{N^i}^i\}$, where $V_i$ stands for the scale of sub-region $R_i$. Considering a constructed sample set $\mathcal{D} = \{x_0^s, \cdots, x_K^s\} \subset X^*$ where $X^* = \mathbb{E}(\mathcal{X})$ denotes expectation input. Denote $V_i^*$ as the scale of expected sub-region $R_i^*$, when $\mathcal{D}$ satisfies $p(x_i^s \in R_j^*) = \frac{V_j^*}{V^*}$ for $\forall x_i^s \in \mathcal{D}$, then $\mathcal{D}$ maximizes the information reflecting $\mathcal{X}$ in expectation.*

We provide the complete proof of Theorem 3.2 in Appendix Sec. A. Theorem 3.2 implies that to construct an effective calibration set we should: (1) partition the data space into meaningful regions (subdomains) and (2) draw samples from each region in proportion to its prevalence. In practical settings where $V_k$ is unknown, a robust strategy is to cluster the dataset into $K$ regions and then sample uniformly inside each cluster (this approximates proportional representation under mild assumptions).

**Observation 2.** *Activations in deeper layers tend to be distinctive, with the majority of samples being highly concentrated, while a few samples are outliers as shown in Fig. 4a.*

For the expected distribution, we prefer representative distribution while the outliers are spiking samples with minimal density. However, when we divide the subdomains and sample within, the selected probability of outliers is increased, which disrupts our expected distribution. Therefore, we propose first to filter the noisy outliers using deep layer statistics for each candidate sample $x_i \in \mathcal{D}$:

$$m_{i,j} := \text{mean}\left(\text{layer}_j(x_i)\right), \qquad s_{i,j} := \text{var}\left(\text{layer}_j(x_i)\right), \quad j \in L, \qquad (9)$$

where $L$ is all used layers union, $D$ is the candidate samples union, $\text{layer}_j(\cdot)$ denotes the activation in $j$-th layer. We then compute a *noise-score* using global robust moments: mean value $m_{i,j}$ and variance $s_{i,j}$:

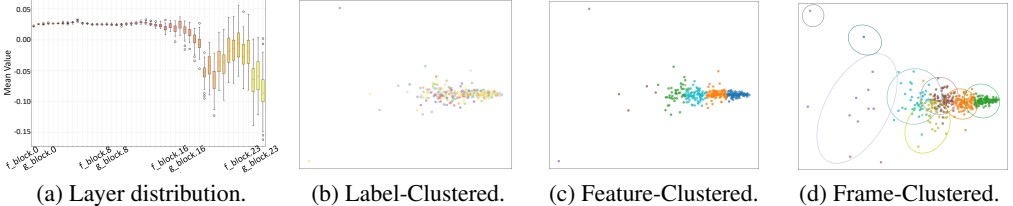

(a) Layer distribution.  (b) Label-Clustered.  (c) Feature-Clustered.  (d) Frame-Clustered.

Figure 4: **The motivation and effect of Noise-Filtered Diverse Sampling.** (a): Layer distribution characteristic of VGGT (Wang et al., 2025a). (b): Visualization of label-based clustered samples, colors denote different labels. (c): Visualization of direct feature-based clustered samples, colors denote clusters. (d): Visualization of frame-wise distribution-based clustered samples, colors denote clusters. **We provide more analysis in Appendix Sec. E.**

$$
\mu_j = \frac{1}{|\mathcal{D}|} \sum_i m_{i,j}, \qquad \sigma_j = \sqrt{\frac{1}{|\mathcal{D}|} \sum_i (m_{i,j} - \mu_j)^2 + \varepsilon},
$$

$$
\nu_j = \frac{1}{|\mathcal{D}|} \sum_i s_{i,j}, \qquad \tau_j = \sqrt{\frac{1}{|\mathcal{D}|} \sum_i (s_{i,j} - \nu_j)^2 + \varepsilon}, \tag{10}
$$

$$
\text{score}(x_i) = \sqrt{\sum_{j \in L} \left( \frac{m_{i,j} - \mu_j}{\sigma_j} \right)^2 + \sum_{j \in L} \left( \frac{s_{i,j} - \nu_j}{\tau_j} \right)^2},
$$

where $\varepsilon$ is a small constant for numerical stability. We then filter out high-noise samples by thresholding the score:

$$
\mathcal{D}_{\text{filtered}} = \{ x_i \in \mathcal{D} \mid \text{score}(x_i) \leq T \}, \tag{11}
$$

where $T$ is set by a percentile (e.g. keep the lowest $p\%$ scores). This filtering keeps samples close to the "typical" distribution and removes outliers that would otherwise skew quantization calibration.

**Observation 3.** *Feature clusters based on raw labels are sub-optimal for visual-geometry tasks.*

We visualized the distribution of different samples and their corresponding labels in Fig. 4b and Fig. 4c. We found that the feature of these samples is highly concentrated and difficult to divide, and using labels directly as classification criteria achieves sub-optimal results. Geometry samples are usually a complex scene containing multiple objects. Therefore, labels often do not directly represent their semantic information. However, we identify that VGGT (Wang et al., 2025a) contains a strong inductive bias: it models the relative relationship between the first frame and subsequent frames. This motivates a structural metric derived from frame-wise features.

Given output feature $\mathbf{A}^i \in \mathbb{R}^{n \times d}$ of sample $x_i$ with $n = s \times f$ (spatial tokens per frame $s$ and $f$ frames). We first reshape $\mathbf{A}^i$ into frame-wise vectors and construct a compact *frame-aware correlation vector* $\mathbf{c}^i \in \mathbb{R}^{f-1}$ by measuring the normalized similarity between the first frame and each subsequent frame:

$$
\mathbf{A}^i \to \widetilde{\mathbf{A}}^i = \left[ \mathbf{a}_0^i, \mathbf{a}_1^i, \ldots, \mathbf{a}_{f-1}^i \right]^\top \in \mathbb{R}^{f \times \hat{d}}, \qquad \hat{d} := s \times d,
$$

$$
c_t^i = \frac{\langle \mathbf{a}_0^i, \mathbf{a}_t^i \rangle}{\|\mathbf{a}_0^i\|_2 \, \|\mathbf{a}_t^i\|_2}, \qquad t = 1, \ldots, f - 1. \tag{12}
$$

We then cluster the set $\{\mathbf{c}^i\}_{x_i \in \mathcal{D}_{\text{filtered}}}$ using K-Means to obtain $K$ regions $\mathcal{R} = \{R_1, \ldots, R_K\}$. According to Theorem 3.2, uniform sampling within each region yields a calibration set that better reflects $\mathcal{X}$. Concretely:

Table 1: Camera Pose Estimation on CO3Dv2 (Reizenstein et al., 2021). **Bold:** The best result.

| Method | Bit-Width (W/A) | frames=10 | | | | frames=20 | | | |
|---|---|---|---|---|---|---|---|---|---|
| | | AUC@30↑ | AUC@15↑ | AUC@5↑ | AUC@3↑ | AUC@30↑ | AUC@15↑ | AUC@5↑ | AUC@3↑ |
| Full Prec. | 16/16 | 89.5 | 83.2 | 66.1 | 54.9 | 90.0 | 83.9 | 67.8 | 56.9 |
| RTN | 8/8 | 88.1 | 80.7 | 60.3 | 46.7 | 88.1 | 80.6 | 60.2 | 46.5 |
| BRECQ | 8/8 | 88.3 | 81.2 | 61.2 | 48.7 | 88.2 | 81.2 | 61.0 | 48.8 |
| QDrop | 8/8 | 88.8 | 81.8 | 61.9 | 49.1 | 88.5 | 81.8 | 61.8 | 49.1 |
| DopQ-ViT | 8/8 | 88.9 | 81.8 | 63.2 | 51.5 | 88.8 | 81.8 | 63.1 | 51.4 |
| GPTQ | 8/8 | 89.1 | 82.6 | 64.0 | 52.1 | 89.1 | 82.6 | 63.2 | 51.5 |
| SmoothQuant | 8/8 | 89.1 | 82.5 | 64.8 | 53.2 | 89.1 | 82.5 | 64.6 | 53.1 |
| QuaRot | 8/8 | **89.4** | 83.0 | 65.9 | 54.6 | 89.4 | 83.0 | **66.0** | 54.7 |
| **QuantVGGT** | 8/8 | **89.4** | **83.1** | **66.1** | **54.8** | **89.6** | **83.2** | **66.0** | **54.9** |
| RTN | 6/6 | 88.1 | 80.1 | 58.1 | 43.7 | 88.1 | 80.2 | 57.6 | 43.1 |
| BRECQ | 6/6 | 88.3 | 80.4 | 58.7 | 43.9 | 88.3 | 80.4 | 58.6 | 43.8 |
| QDrop | 6/6 | 88.5 | 80.8 | 58.9 | 44.1 | 88.4 | 80.6 | 58.8 | 43.9 |
| DopQ-ViT | 6/6 | 88.5 | 80.7 | 59.4 | 44.8 | 88.5 | 80.7 | 59.4 | 44.7 |
| GPTQ | 6/6 | 88.7 | 81.0 | 61.3 | 46.3 | 88.7 | 81.0 | 61.2 | 46.2 |
| SmoothQuant | 6/6 | 88.8 | 81.3 | 61.4 | 47.4 | 88.9 | 81.5 | 61.5 | 47.5 |
| QuaRot | 6/6 | 89.0 | 81.8 | 62.5 | 49.4 | 89.1 | 81.9 | 62.6 | 49.5 |
| **QuantVGGT** | 6/6 | **89.2** | **82.7** | **65.2** | **53.8** | **89.3** | **82.8** | **65.6** | **54.1** |
| RTN | 4/4 | 77.7 | 63.9 | 31.4 | 16.6 | 75.8 | 60.7 | 26.5 | 12.8 |
| BRECQ | 4/4 | 78.8 | 65.2 | 34.3 | 20.1 | 78.8 | 65.3 | 34.1 | 20.0 |
| QDrop | 4/4 | 79.1 | 66.7 | 35.7 | 22.0 | 79.2 | 66.7 | 35.6 | 21.9 |
| DopQ-ViT | 4/4 | 80.3 | 68.3 | 38.3 | 23.3 | 80.4 | 68.4 | 35.5 | 22.0 |
| RepQ-ViT | 4/4 | 82.4 | 69.9 | 37.1 | 20.2 | 81.4 | 68.1 | 34.8 | 18.4 |
| GPTQ | 4/4 | 80.5 | 68.6 | 38.7 | 23.2 | 80.6 | 68.7 | 35.6 | 22.1 |
| AWQ | 4/4 | 81.2 | 69.9 | 40.8 | 26.5 | 80.2 | 68.7 | 38.9 | 24.8 |
| SmoothQuant | 4/4 | 75.4 | 60.1 | 25.8 | 12.3 | 75.4 | 60.1 | 25.5 | 12.1 |
| QuaRot | 4/4 | 81.8 | 70.3 | 40.1 | 23.5 | 81.6 | 69.8 | 39.4 | 22.6 |
| **QuantVGGT** | 4/4 | **86.9** | **78.7** | **57.3** | **43.6** | **88.2** | **80.2** | **58.9** | **45.1** |

$$R_k = \{x_i \in \mathcal{D}_{\text{filtered}} \mid \hat{y}_i = k\}, \qquad \mathcal{D}_{\text{calib}} = \bigcup_{k=1}^{K} \Omega(R_k), \qquad (13)$$

where $\hat{y}_i$ is the cluster assignment and $\Omega(\cdot)$ denotes a uniform sampler. This Noise-Filtered Diverse Sampling pipeline reduces the influence of noisy outliers, leverages VGGT's frame-relative inductive bias to form semantically meaningful clusters as shown in Fig. 4d, and yields a calibration set that better approximates the true data distribution for PTQ.

## 4 EXPERIMENTS

### 4.1 EXPERIMENTAL AND EVALUATION SETTINGS

**Evaluation Settings.** We select VGGT-1B (Wang et al., 2025a) as our base model and conduct all the quantization experiments on it. To validate the effectiveness of our proposed method, we conduct camera pose estimation experiments on CO3Dv2 (Reizenstein et al., 2021) and point map estimation experiments on DTU (Jensen et al., 2014). For the quantization setting, we select two of the most widely studied bit settings W8A8 (8-bit weight and 8-bit activation quantization) and W4A4, as they have better hardware adaptability and bring more acceleration and compression effects (Xiao et al., 2023; Ashkboos et al., 2024). **More details can be found in Appendix Sec. B.**

**Baseline Methods.** For quantization baseline methods, we adopt the commonly used PTQ baseline Round-To-Nearest (RTN), BRECQ (Li et al., 2021), and QDrop (Wei et al., 2022). For 2D-vision transformer baseline, we select strong DopQ-ViT (Yang et al., 2024) and RepQ-ViT (Li et al., 2023). For language transformer baseline, we select strong GPTQ (Frantar et al., 2022), AWQ (Lin et al., 2024), SmoothQuant (Xiao et al., 2023), and QuaRot (Ashkboos et al., 2024).

### 4.2 MAIN RESULTS

**Camera Pose Estimation.** We conduct camera pose estimation experiments using VGGT-1B (Wang et al., 2025a) on CO3Dv2 dataset (Reizenstein et al., 2021). Following prior works (Wang et al., 2025a), we randomly sample 10 frames for evaluation and further expand to 20 frames for a more

generalized evaluation. The results are presented in Tab. 1. Under the relatively simpler W8A8 setting, most quantization methods can maintain relatively good performance but inevitably experience certain performance degradation. Quantvggt preserves 99.9% performance under W8A8, with AUC@30 of 89.4 and 89.5 for FP (Full Precision). For the more aggressive W4A4 setting, all quantization methods showed significant performance degradation, such as current SOTA method QuaRot (Ashkboos et al., 2024) only achieving 81.6 AUC@30 under 20 frames. While, QuantVGGT still achieved 88.2, maintaining 98% of the model's performance. QuantVGGT can achieve significant performance improvements even under extreme quantization settings compared to existing methods, demonstrating its quantization friendliness towards 3D reconstruction models.

**Point Map Estimation.** To comprehensively evaluate the generalized quantization performance of VGGT, we further extend the experiment to the point map estimation task on DTU dataset (Jensen et al., 2014). For evaluation, we sample keyframes every 5 images. The results are presented in Tab. 2. It is worth mentioning that the calibration dataset is all from CO3Dv2 training set, meaning that the DTU data are unknown for the calibration process. However, QuantVGGT still generalizes well on the point map estimation task, with even improved metrics compared with the FP model under W8A8. For W4A4 setting, all existing methods show notable performance degradation, like QuaRot with ACC. of only 1.593. While QuantVGGT achieves ACC. of 1.282, significantly closer to the FP performance of 1.185. This demonstrates QuantVGGT's ability to adapt to large 3D models such as VGGT quantization, and can maintain strong generalization ability with an efficient PTQ process.

Table 2: Point Map Estimation on DTU (Jensen et al., 2014).

| Method | Bit-Width (W/A) | Acc.↓ | | Comp.↓ | | N.C.↑ | |
|---|---|---|---|---|---|---|---|
| | | Mean | Med. | Mean | Med. | Mean | Med. |
| Full Prec. | 16/16 | 1.185 | 0.714 | 2.232 | 1.313 | 0.694 | 0.779 |
| RTN | 8/8 | 1.216 | 0.730 | 2.237 | 1.310 | 0.687 | 0.773 |
| BRECQ | 8/8 | 1.212 | 0.725 | 2.236 | 1.292 | 0.690 | 0.774 |
| QDrop | 8/8 | 1.204 | 0.720 | 2.239 | 1.297 | 0.692 | 0.780 |
| DopQ-ViT | 8/8 | 1.200 | 0.712 | 2.235 | 1.290 | 0.691 | 0.783 |
| GPTQ | 8/8 | 1.196 | 0.721 | 2.226 | 1.303 | 0.688 | 0.778 |
| SmoothQuant | 8/8 | 1.201 | 0.719 | 2.229 | 1.281 | 0.692 | 0.776 |
| QuaRot | 8/8 | 1.184 | 0.712 | 2.231 | 1.311 | 0.694 | 0.778 |
| **QuantVGGT** | 8/8 | **1.182** | **0.710** | **2.215** | **1.276** | **0.700** | **0.788** |
| RTN | 4/4 | 1.700 | 0.930 | 2.028 | 1.099 | 0.656 | 0.757 |
| BRECQ | 4/4 | 1.688 | 0.924 | 2.024 | 1.082 | 0.662 | 0.762 |
| QDrop | 4/4 | 1.642 | 0.912 | 2.012 | 1.073 | 0.673 | 0.754 |
| DopQ-ViT | 4/4 | 1.587 | 0.906 | 2.003 | 1.075 | 0.673 | 0.755 |
| GPTQ | 4/4 | 1.442 | 0.899 | 1.997 | 1.068 | 0.675 | 0.761 |
| SmoothQuant | 4/4 | 1.740 | 0.944 | 1.993 | 1.083 | 0.675 | 0.764 |
| QuaRot | 4/4 | 1.593 | 0.916 | 2.034 | 1.096 | 0.670 | 0.757 |
| **QuantVGGT** | 4/4 | **1.282** | **0.743** | **1.992** | **1.068** | **0.681** | **0.774** |

**Point Cloud Reconstruction.** To further demonstrate the generalization ability of QuantVGGT beyond Co3Dv2 and DTU, we conduct additional point cloud reconstruction experiments on the 7-Scenes (Shotton et al., 2013) and NRGBD (Azinović et al., 2022b) datasets under the same W4A4 quantization protocol. The results are in Tab. 3. On both datasets, QuantVGGT substantially outperforms current SOTA methods SmoothQuant and QuaRot, also maintains nearly lossless performance. These results demonstrate that QuantVGGT retains strong reconstruction quality across diverse benchmarks, highlighting its robustness and generality.

Table 3: Point Cloud Reconstruction experiment on VGGT (Wang et al., 2025a) using 7-Scenes (Shotton et al., 2013) and NRGBD (Azinović et al., 2022b) dataset.

| Method | Acc.↓ | | Comp.↓ | | N.C.↑ | |
|---|---|---|---|---|---|---|
| | Mean | Med. | Mean | Med. | Mean | Med. |
| 7-Scenes | | | | | | |
| Full Prec. | 0.025 | 0.013 | 0.036 | 0.020 | 0.728 | 0.836 |
| SmoothQuant | 0.370 | 0.261 | 0.498 | 0.361 | 0.484 | 0.477 |
| QuaRot | 0.030 | 0.016 | 0.042 | 0.022 | 0.701 | 0.800 |
| **QuantVGGT** | **0.026** | **0.013** | **0.037** | **0.019** | **0.718** | **0.812** |
| NRGBD | | | | | | |
| Full Prec. | 0.015 | 0.009 | 0.017 | 0.007 | 0.878 | 0.969 |
| SmoothQuant | 0.479 | 0.393 | 0.614 | 0.489 | 0.515 | 0.513 |
| QuaRot | 0.034 | 0.021 | 0.030 | 0.015 | 0.820 | 0.948 |
| **QuantVGGT** | **0.019** | **0.013** | **0.021** | **0.010** | **0.850** | **0.959** |

### 4.3 ABLATION STUDY

To validate the effectiveness of each proposed component, we conduct an ablation study. All the experiments are conducted under W4A4 quantization setting on CO3Dv2 (Reizenstein et al., 2021).

**Quantization Architecture.** We first validate the proposed *Dual-Smoothed Fine-Grained Quantization* (DSFQ) and present the result in Tab. 4. We denote naive quantization without any smoothing as *Base*. We further compare with the rotation-only (*Rotation*) and scale-only (*Scale*) methods with our proposed DSFQ. Naive quantization shows significant performance collapse with AUC@3 of

only 9.7. While scale-based and rotation-based methods further smoothed data distribution and showed certain improvement, they still exhibit inevitable degradation.

While our proposed DSFQ combines the advantages of both rotation and scale and utilizes fine-grained quantization granularity, greatly preserves the performance. Furthermore, to validate the necessity of our pre-global-rotation and post-local-smooth (*Rot.-Scale*), we also compared with pre-smooth and post-rotation (*DSFQ*). In terms of performance, our *DSFQ* approach has indeed achieved significant improvements compared to *Scale-Rot.*. This

Table 4: Ablation study on quantization architecture.

| Method | AUC@30$_\uparrow$ | AUC@15$_\uparrow$ | AUC@5$_\uparrow$ | AUC@3$_\uparrow$ |
|---|---|---|---|---|
| Full Prec. | 89.5 | 83.2 | 66.1 | 54.9 |
| Base | 76.9 | 61.5 | 23.9 | 9.7 |
| *Rotation* | 83.6 | 72.3 | 46.3 | 32.5 |
| *Scale* | 81.9 | 70.1 | 38.5 | 21.2 |
| *Scale-Rot.* | 86.7 | 78.5 | 56.8 | 42.9 |
| **DSFQ** | **86.9**$_{+3.3}$ | **78.7**$_{+6.4}$ | **57.3**$_{+11.0}$ | **43.6**$_{+11.1}$ |

proves that the effect of smoothing the rotated space is more stable, while smoothing first and then rotating will weaken the influence of smoothing due to its rearrangement of outliers between channels, demonstrating the effectiveness of our method.

**Sampling Strategy.** We then validate the proposed *Noise-Filtered Diverse Sampling* (NFDS) and show the result in Fig. 5. We denote the naive random sampling strategy as *Random*. We then compare with random sampling from outlier-filtered dataset (*Filtered*) and sampling from frame-based clustered dataset without filtering (*Clustered*) strategy. All the results are conducted under five different random seeds. We present the mean performance and its corresponding variance in the bar plot. Random selection not only fails to guarantee diversity but also results in significant variance due to the influence of outliers. The filtered data quality was improved, and the variance was significantly reduced. Our clustering method

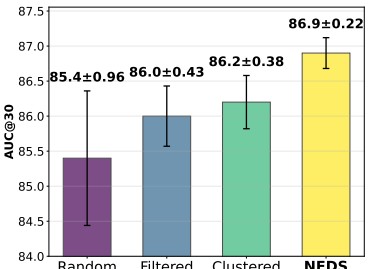

Figure 5: Sample strategy ablation.

significantly enhances diversity and improves average performance, but there is still variance due to the presence of outliers. The final combined NFDS ensures both the removal of outliers and well diversity, ensuring average performance while being more stable.

## 4.4 EFFICIENCY ANALYSIS

To verify the deployment efficiency of quantized VGGT, we report the hardware latency in Fig. 6. All the experiments are conducted on a single NVIDIA RTX 4090 24G GPU with CUDA 12.4. We use CUT-LASS (Thakkar et al., 2023) on top of PyTorch for performing low-bit INT matrix multiplication. Compared with naive quantization without any smooth techniques, our dual-smoothed fine-grained quantization only brings additional 0.2% latency cost at W4A4, while significantly preserving the quantized model performance. Our W4A4 QuantVGGT even surpass the naive W8A8 performance, and with a significant performance gap of naive W4A4. This indicates that our VGGT-specialized quantization scheme greatly outperforms

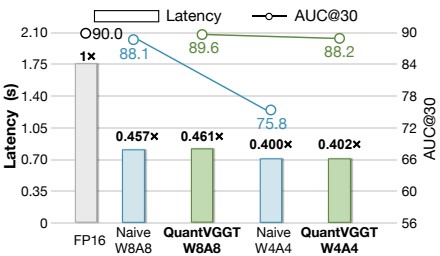

Figure 6: Latency speedup and corresponding performance under 20 frames.

the existing naive quantization with little extra burden.

## 5 CONCLUSION

In this paper, we propose the first Post-Training Quantization (PTQ) framework for VGGTs, namely QuantVGGT. Specifically, we identify the quantization-unfriendly distribution brought by data-independent tokens and the highly unstable calibration dataset inherent in 3D multi-view data. We then propose Dual-Smoothed Fine-Grained Quantization to smooth the heavy-tailed distribution. We also design Noise-Filtered Diverse Sampling to constructs frame-aware diverse calibration clusters to ensure stable dataset. Extensive experiments demonstrate that QuantVGGT achieves state-of-the-art performance under different bit-widths and greatly surpasses existing quantization methods.

## 6 ACKNOWLEDGEMENTS

This work is supported by the National Natural Science Foundation of China under Grant Number 62476264 and 62406312, the Beijing Natural Science Foundation under Grant Number 4244098, the Science Foundation of the Chinese Academy of Sciences, and the Swiss National Science Foundation (SNSF) project 200021E_219943 Neuromorphic Attention Models for Event Data (NAMED).

## 7 ETHICS STATEMENT

This research strictly adheres to the ICLR Code of Ethics with no ethics-related risks: it uses public open-source models (VGGT (Wang et al., 2025a)) and focuses on algorithmic innovation for inference acceleration and compression, without involving scenarios endangering public safety, infringing privacy, or producing discrimination.

## 8 REPRODUCIBILITY STATEMENT

To ensure reproducibility, experimental configurations, method details, and evaluation metrics are thoroughly described in Sec. 4.1 and Appendix Sec. B. Experimental results of comparative methods are sourced from public literature, and our experiments strictly follow the same configurations as baseline methods for fair comparison. Complete source code for reproducing results will be publicly released upon paper publication. The raw reconstruction files are attached in the supplementary materials. For the theorem used in the paper, we also provided a detailed proof in Appendix Sec. A.

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

## A  PROOF OF THEOREM 3.2

*Proof.* For $X^* = \mathbb{E}(\mathcal{X})$, we have $N^* = \max\{N^i\}$. Since $N^i$ is finite, $N^*$ is also finite. And for the scale $V_i^*$ of sub-region $R_i^*$ of $X^*$, we have $V_i^* = \mathbb{E}_j \mathbb{E}_i(V_i^j)$, then we have $V_j^* = V_i^*$ for $\forall i, j$. Therefore, $V_j^* = V^*/N^*$.

Consider $\mathcal{D} = \{x_0, \cdots, x_K\} \subset X^*$, to maximize the information of $\mathcal{D}$, we need to maximize:

$$\max \mathcal{H}(\mathcal{D}) = -\sum_{i=0}^{K} \sum_{j=1}^{N^*} p(x_i \in R_j^*) \log p(x_i \in R_j^*). \tag{14}$$

Since all samples are independent, Eq. 14 is equivalent to

$$\max \sum_i \mathcal{H}_i(\mathcal{D}) = \sum_{i=0}^{K} \left( \max - \sum_{j=1}^{N^*} p(x_i \in R_j^*) \log p(x_i \in R_j^*) \right). \tag{15}$$

Without loss of generality, we only need to discuss $\max \mathcal{H}_i(\mathcal{D})$. To simplify writing, we denote $p(x_i \in R_j^*)$ as $p_j$, and the above problem can be derived as:

$$\max \mathcal{H}_i(\mathcal{D}) = \max - \sum_{j=1}^{N^*} p_j \log p_j. \tag{16}$$

To solve this problem, we introduce Lagrangian multiplier $\lambda$ and construct Lagrangian as:

$$\mathcal{L}(p_1, \cdots, p_{K^*}, \lambda) = -\sum_{j=1}^{N^*} p_j \log p_j + \lambda \left( \sum_{j=1}^{N^*} p_j - 1 \right). \tag{17}$$

We can solve this by letting:

$$\frac{\partial \mathcal{L}}{\partial p_j} = 0, \ \sum_{j=1}^{N^*} p_j = 1. \tag{18}$$

Therefore, we have:

$$\frac{\partial \mathcal{L}}{\partial p_j} = -(\log p_j + 1) + \lambda = 0 \implies p_j = e^{\lambda - 1}. \tag{19}$$

Substitute into $\sum_{j=1}^{K^*} p_j = 1$:

$$N^* e^{\lambda - 1} = 1 \implies e^{\lambda - 1} = \frac{1}{N^*} \implies p_j = \frac{1}{N^*}. \tag{20}$$

At this point, $\mathcal{H}_i(x^s) = \log N^*$ (maximum entropy).

Given that $\forall j, V_j^* = V^*/N^*$, when $p_j = \frac{V_j^*}{V^*} = \frac{1}{N^*}$, the information is maximized.

Therefore, Theorem 3.2 holds.

$\square$

## B EXPERIMENT SETTINGS

For camera pose estimation, we randomly select 10 frames and 20 frames to validate the performance under varying sequence lengths following (Wang et al., 2025a; 2023). We use the standard metric AUC (Wang et al., 2023), which combines RRA (Relative Rotation Accuracy) and RTA (Relative Translation Accuracy). For point map estimation, we sample keyframes every 5 images. Consistent with prior works (Azinović et al., 2022a; Wang et al., 2024), we use Accuracy (Acc.), Completion (Comp.), and Normal Consistency (N.C.) to validate the performance.

Same with prior works (Li et al., 2021; Xiao et al., 2023), we adopt channel-wise weight quantization strategy. For *Fine-Grained Quantization Granularity*, we further use dynamic token-wise quantization for activation. We use symmetry quantization for both weight and quantization for efficiency. For Hadamard rotation, we use random Hadamard matrix following (Ashkboos et al., 2024; Tseng et al., 2024). For the calibration process, we use block-wise quantization pipeline following (Li et al., 2021; Wei et al., 2022).

For hyperparameter setting, we set $\alpha = 0.5$ in Eq. 7 for channel-wise scale initialization. We set $p = 0.2$ in Eq. 11 and cluster center $K = 8$ in Eq. 13 for calibration dataset construction. We select 40 samples from a total 400 samples pool. During calibration process, we set channel-wise scale $\hat{c}$ and quantization parameters $\Delta$ as learnable. We set the learning rate as $5e^{-3}$ for $\hat{c}$ and $5e^{-2}$ for $\Delta$.

Table 5: More quantization setting results on CO3Dv2 (Reizenstein et al., 2021) under W4A4.

| Method | AUC@30↑ | AUC@15↑ | AUC@5↑ | AUC@3↑ |
|---|---|---|---|---|
| Full Prec. | 90.5 | 84.4 | 67.9 | 57.0 |
| QuaRot | 82.8 | 71.4 | 41.0 | 24.4 |
| **QuantVGGT** | 87.9 | 79.0 | 58.4 | 44.7 |
| **QuantVGGT** *(Mixed Precision)* | 88.0 | 78.8 | 58.4 | 44.6 |
| **QuantVGGT** *(Asymmetric)* | 87.8 | 78.9 | 58.6 | 44.8 |
| **QuantVGGT** *(QAT)* | 88.1 | 80.2 | 58.6 | 44.6 |

Table 6: Point Cloud Reconstruction experiment on Fast3R (Yang et al., 2025) using 7-Scenes (Shotton et al., 2013) dataset.

| Method | Acc.↓ | | Comp.↓ | | N.C.↑ | |
|---|---|---|---|---|---|---|
| | Mean | Med. | Mean | Med. | Mean | Med. |
| Full Prec. | 0.049 | 0.021 | 0.069 | 0.021 | 0.624 | 0.683 |
| SmoothQuant | 0.497 | 0.448 | 0.319 | 0.244 | 0.586 | 0.625 |
| QuaRot | 0.312 | 0.258 | 0.149 | 0.080 | 0.593 | 0.637 |
| **QuantVGGT** | **0.049** | **0.022** | **0.070** | **0.022** | **0.624** | **0.683** |

Table 7: Camera Pose Estimation comparison with other acceleration methods on CO3Dv2 (Reizenstein et al., 2021)

| Method | FLOPs (T)↓ | AUC@30↑ | AUC@15↑ | AUC@5↑ | AUC@3↑ |
|---|---|---|---|---|---|
| VGGT | 5.84 | 89.5 | 83.2 | 66.1 | 54.9 |
| FastVGGT | 1.68 | 82.7 | 71.3 | 39.2 | 25.1 |
| **QuantVGGT** | **1.40** | **86.9** | **78.7** | **57.3** | **43.6** |

## C MORE EXPERIMENTS RESULTS

We further evaluate QuantVGGT under mixed precision and asymmetric quantization under W4A4 on CO3Dv2. We use the first 200 sequences of each class for faster evaluation. Results are present in Tab. 5. These results indicate that QuantVGGT is not tied to a specific quantization format and

Table 8: Ablation study on activation quantization granularity under W4A4.

| Compute | Granularity | Memory Opt.↑ | Latency Opt.↑ | AUC@30↑ |
|---------|-------------|--------------|---------------|---------|
| Full Prec. | Full Prec. | 1.00× | 1.00× | 89.5 |
| Static | Tensor-wise | 3.65× | 2.50× | 82.2 |
| Static | Token-wise | 3.65× | 2.50× | 84.1 |
| Dynamic | Tensor-wise | 3.65× | 2.49× | 82.7 |
| Dynamic | Token-wise | 3.65× | 2.49× | **86.9**$_{+2.8}$ |

can be readily deployed under mixed-precision or asymmetric settings while maintaining strong accuracy. Due to the limited computation resources but to still provide empirical evidence, we conduct a lightweight QAT experiment by fine-tuning the quantized model for 1 epoch on the same calibration dataset used by QuantVGGT (i.e., no extra data are introduced). The results are in Tab. 5. QAT indeed improves low-bit performance, but the computational overhead is substantial.

To validate QuantVGGT generality across architectures, we additionally evaluate QuantVGGT on Fast3R (Yang et al., 2025), a recently proposed transformer-based 3D reconstruction backbone. We report point-cloud reconstruction results on 7-Scenes (Shotton et al., 2013) dataset in Tab. 6. QuantVGGT achieves practically lossless performance relative to the full-precision Fast3R model and substantially surpasses strong quantization baselines (QuaRot and SmoothQuant) on all metrics.

We further provide a comparison with other VGGT (Wang et al., 2025a) acceleration methods. FastVGGT (Shen et al., 2025) accelerates VGGT via token compression/token merging, while our QuantVGGT focuses on post-training quantization. These techniques are conceptually orthogonal and, in principle, can be combined. To provide a concrete comparison, we evaluate runtime FLOPs and reconstruction accuracy on Co3Dv2 in Tab. 7. From these results, we observe that: QuantVGGT achieves larger FLOPs reduction than FastVGGT (1.40T vs. 1.68T), while also maintaining significantly higher accuracy at all metrics. This highlights that quantization-based acceleration (QuantVGGT) can be highly competitive with token-compression-based acceleration (FastVGGT), even though they target different aspects of the model.

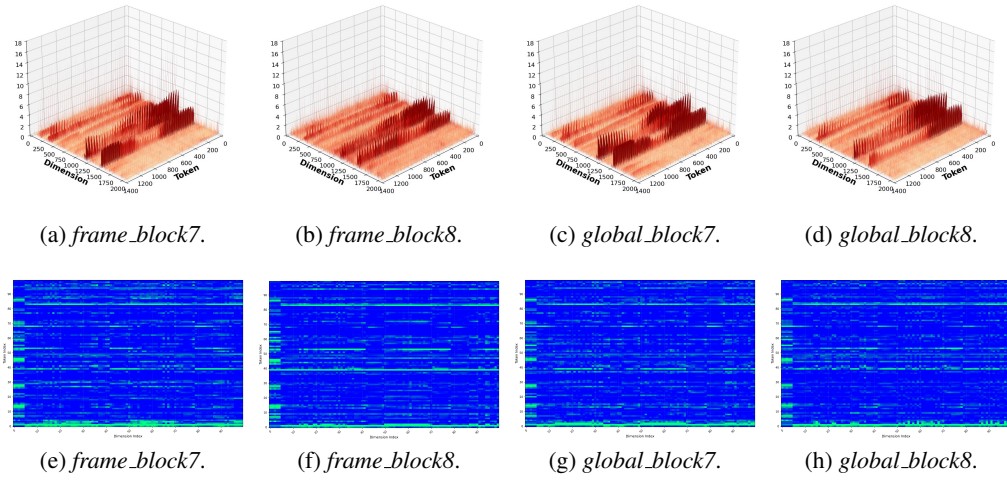

(a) *frame_block7.*   (b) *frame_block8.*   (c) *global_block7.*   (d) *global_block8.*

(e) *frame_block7.*   (f) *frame_block8.*   (g) *global_block7.*   (h) *global_block8.*

Figure 7: More salient distribution and registered tokens saliency.

## D  MORE ANALYSIS ON DUAL-SMOOTHED FINE-GRAINED QUANTIZATION

In this section, we provide more analysis on the proposed *Dual-Smoothed Fine-Grained Quantization (DSFQ)*.

We first present more visualizations of the heavy-tailed activation distribution of VGGT (Wang et al., 2025a) and the amplified salient phenomenon brought by data-independent tokens in Fig. 7. It can be seen that this salient phenomenon is reflected in different layers of VGGT, which will bring universal bottlenecks to the quantization performance. And these data-independent tokens almost always have more severe salient phenomena than other adjacent image tokens, which reflects the salient amplified effect of these special registration tokens.

Also, to verify the trade-off between performance and latency in our fine-grained quantization, we tested the impact of different quantization granularities on latency and performance, and present the results in Tab. 8. For weight quantization, we apply uniform channel-wise quantization. Here, we only study activation quantization granularity. 'Static' compute denotes computing the quantization parameter $\Delta$ in Eq. 3 during calibration and fixed during inference. 'Dynamic' denotes computing $\Delta$ during inference based on min-max value. It can be seen that dynamic quantization and token-wise quantization impose almost no burden on memory and only result in an additional latency of 0.01% when used simultaneously. However, this fine-grained quantization brings significant performance improvements. Compared to static tensor-wise quantization, dynamic token-wise quantization only brings 0.01% additional latency but improves AUC@30 from 82.2 to 86.9.

We also validate the proposed scaling and Hadamard-based rotation inference overload under W4A4. The results are shown in Tab. 9. For the Hadamard transform, we follow existing work QuaRot (Ashkboos et al., 2024) that apply fast-hadamard-transform with CUDA kernel for deployment. For the smooth factor, we follow SmoothQuant (Xiao et al., 2023) that fuses the factor within previous LayerNorm operation, which introduces no extra inference burden. With fast-hadamard-transform CUDA kernel, the Hadamard transform almost introduce no extra burden but significantly improves model performance form 81.9 AUC@30 to 86.9. This proves both the efficiency and effectiveness of the proposed DSFQ pipeline.

Table 9: Inference overload of proposed quantization preprocess operations under W4A4.

| Method | Memory Opt.↑ | Latency Opt.↑ | AUC@30↑ |
|---|---|---|---|
| QuantVGGT (*w/o* Hadamard) | 3.65× | 2.50× | 81.9× |
| QuantVGGT | 3.65× | 2.49× | 86.9× |

We further conduct an ablation study on Hadamard matrix and smooth factor. For Hadamard matrix, as it is randomly generated, we use different random seed for ablation. And we vary smoothing $\alpha$ for sensitivity analysis. Here, we present the results in Tab. 10. These results indicate that QuantVGGT hyperparameter is not rely on heavily tuning and is robust to small perturbations.

Table 10: Ablation study of Hadamard and smooth factor sensitivity on CO3Dv2 (Reizenstein et al., 2021) under W4A4.

| Method | AUC@30↑ | AUC@15↑ | AUC@5↑ | AUC@3↑ |
|---|---|---|---|---|
| Full Prec. | 89.5 | 83.2 | 66.1 | 54.9 |
| QuaRot | 81.8 | 70.3 | 40.1 | 23.5 |
| **QuantVGGT** (*seed 1, $\alpha = 0.5$*) | 86.9 | 78.7 | 57.3 | 43.6 |
| **QuantVGGT** (*seed 1, $\alpha = 0.3$*) | 86.6 | 78.4 | 57.0 | 43.3 |
| **QuantVGGT** (*seed 1, $\alpha = 0.7$*) | 86.5 | 78.3 | 57.0 | 43.2 |
| **QuantVGGT** (*seed 2, $\alpha = 0.5$*) | 86.8 | 78.6 | 57.1 | 43.5 |
| **QuantVGGT** (*seed 3, $\alpha = 0.5$*) | 86.9 | 78.7 | 57.3 | 43.7 |

# E   MORE ANALYSIS ON NOISE-FILTERED DIVERSE SAMPLING

In this section, we provide more analysis of our proposed *Noise-Filtered Diverse Sampling (NFDS)*.

We conduct ablation study on threshold $p$, cluster number $K$, and calibration sample size $N$. The results (W4A4 QuantVGGT on Co3Dv2, 10 frames) are summarized in Tab. 11. These results show that QuantVGGT is robust over a wide range of hyperparameters, and performance steadily improves with more calibration samples.

Table 11: Ablation study of NFDS hyperparameters on CO3Dv2 (Reizenstein et al., 2021) under W4A4.

| Method | AUC@30↑ | AUC@15↑ | AUC@5↑ | AUC@3↑ |
|---|---|---|---|---|
| Full Prec. | 90.5 | 84.4 | 67.9 | 57.0 |
| QuaRot | 82.8 | 71.4 | 41.0 | 24.4 |
| Calibration Data Size | | | | |
| **QuantVGGT** *(5 samples)* | 86.4 | 77.6 | 53.8 | 38.9 |
| **QuantVGGT** *(10 samples)* | 86.9 | 78.4 | 55.3 | 40.8 |
| **QuantVGGT** *(20 samples)* | 86.9 | 78.4 | 55.3 | 40.8 |
| **QuantVGGT** *(40 samples)* | 87.9 | 79.0 | 58.4 | 44.7 |
| Filter Threshold $p$ | | | | |
| **QuantVGGT** $(p = 0.1)$ | 87.5 | 78.7 | 57.8 | 44.2 |
| **QuantVGGT** $(p = 0.2)$ | 87.9 | 79.0 | 58.4 | 44.7 |
| **QuantVGGT** $(p = 0.3)$ | 87.5 | 78.8 | 57.8 | 44.3 |
| Cluster Number $K$ | | | | |
| **QuantVGGT** $(K = 6)$ | 87.7 | 79.7 | 58.1 | 44.4 |
| **QuantVGGT** $(K = 8)$ | 87.9 | 79.0 | 58.4 | 44.7 |
| **QuantVGGT** $(K = 10)$ | 87.7 | 79.7 | 57.9 | 44.0 |

We present quantization performance using different cluster strategies using five random seeds. We denote our NFDS as *Frame-based*, directly using features to cluster as *Feature-based*, and using prior labels to cluster as *Label-based*. The performance comparison is presented in Fig. 8. It can be seen that the experimental results are consistent with our previous analysis. The dataset constructed based on inductive bias clustering results can bring better average performance and significantly reduce the impact of randomness. However, datasets constructed using other prior knowledge have only slight improvements compared to completely random ones.

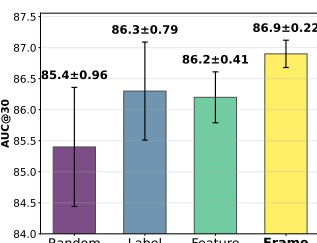

Figure 8: More ablation study on sample strategy.

To evaluate NFDS under extreme cases, we construct a subset from CO3Dv2 (Reizenstein et al., 2021) dataset where the first frame is significantly inconsistent with the rest of the sequence. Specifically, we filter calibration candidates whose average cosine similarity between the first frame and all other frames is $< 0.1$. On this challenging subset, we compare Random sampling and NFDS using 20 samples and present the results in Tab. 12. NFDS consistently outperforms random sampling because: Outlier filtering removes extreme low-quality samples before clustering; Frame-aware clustering is not solely dependent on the first frame, since clustering focuses on relative inter-frame geometry, rather than absolute appearance quality.

Table 12: Ablation study of calibration data sampling strategy on CO3Dv2 (Reizenstein et al., 2021) extreme cases under W4A4.

| Method | AUC@30↑ | AUC@15↑ | AUC@5↑ | AUC@3↑ |
|---|---|---|---|---|
| Full Prec. | 90.5 | 84.4 | 67.9 | 57.0 |
| Random | 87.7 | 78.6 | 57.6 | 43.4 |
| **NFDS** | 87.9 | 79.0 | 58.4 | 44.7 |

## F    CALIBRATION AND INFERENCE COMPUTATION RESOURCE

In this section, we report the calibration and inference computation resource and the additional burden brought by our proposed methods. We first present the inference comparison in Tab. 14 We present the detailed results in Tab. 13 and the performance are under W4A4. The filter and cluster

Table 13: Ablation study on calibration costs.

| Method | Calibration Overload | | Performance | |
|---|---|---|---|---|
| | GPU Memory (GB)↓ | GPU Time (Hours)↓ | AUC@30↑ | AUC@15↑ |
| Full Prec. | - | - | 90.0 | 83.9 |
| Naive PTQ | 14.4 | 2.53 | 78.8 | 65.2 |
| **QuantVGGT** *w/o filter* | $14.6_{+0.2}$ | $2.56_{+0.03}$ | $86.2_{+7.4}$ | $77.1_{+11.9}$ |
| **QuantVGGT** *w/o cluster* | $14.6_{+0.2}$ | $2.64_{+0.11}$ | $86.0_{+7.2}$ | $76.5_{+11.3}$ |
| **QuantVGGT** | $14.6_{+0.2}$ | $2.67_{+0.14}$ | $86.9_{+9.6}$ | $78.7_{+14.9}$ |

are used in calibration dataset construction and we select 40 samples from 400 data pool to ensure the robustness. Compared to the baseline PTQ process (Li et al., 2021), QuantVGGT only brings an additional memory consumption of 0.02GB and an additional calibration time of 0.1 hour, but brings significant performance improvement. And the additional calibration time is almost only affected by the construction of the calibration dataset. But even our complete sampling strategy only brings an additional 0.14 hours of time, and the total PTQ process only takes 2.67 hours and can be performed on consumer GPUs such as RTX4090. This fully demonstrates that our PTQ algorithm is highly efficient and effective.

Table 14: Efficiency comparison across memory and latency under different bit-width.

| Bit-width (W/A) | Memory Opt.↑ | Latency Opt.↑ |
|---|---|---|
| 16/16 | 1.00× | 1.00× |
| 8/8 (naive) | 1.94× | 2.19× |
| 8/8 (ours) | 1.93× | 2.17× |
| 4/4 (ours) | 3.65× | 2.49× |

To better illustrate the practical acceleration benefits of our QuantVGGT under varying computational demands, we have conducted an additional ablation experiment measuring the inference acceleration of QuantVGGT (W4A4) at different sequence lengths. The detailed latency optimization is shown in Tab. 15. The results show that even as sequence length increases, QuantVGGT consistently provides substantial acceleration with stable scaling trends.

Table 15: Latency optimization of W4A4 QuantVGGT under different input sequence lengths.

| Length | 10 | 20 | 40 | 80 |
|---|---|---|---|---|
| **Latency Optimization** | 2.47× | 2.49× | 2.53× | 2.55× |

# G  THE USE OF LARGE LANGUAGE MODELS (LLMS)

In this paper, Large Language Models are only used as general-purpose auxiliary tools, primarily for document-level auxiliary tasks such as grammar checking and expression refinement. LLMs did not participate in the core conceptualization, method derivation, or experimental design of this research, nor did they contribute to any core writing content.

# H  MORE PERFORMANCE VISUALIZATION

Here, we present more visual comparison results between Full Precision model and W4A4 QuantVGGT in Fig. 9 and Fig. 10.

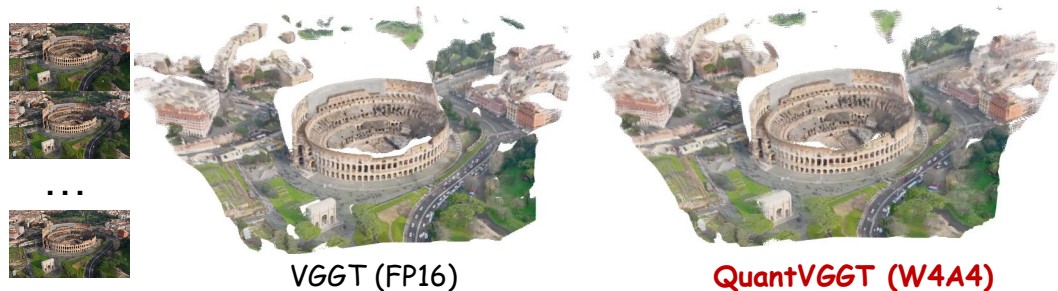

Figure 9: Visual comparison results.

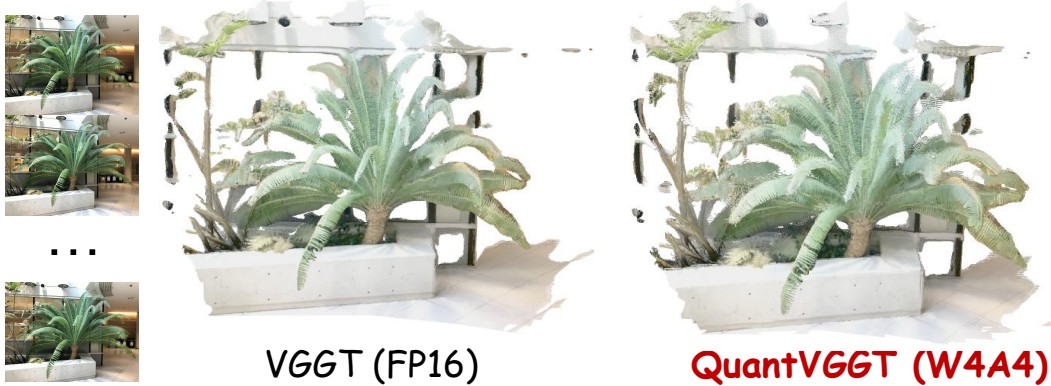

Figure 10: Visual comparison results.

To further demonstrate the effectiveness of our quantized model, we provide additional comparison with RTN and QuaRot (Ashkboos et al., 2024) in Fig. 11, Fig. 12, and Fig. 13.

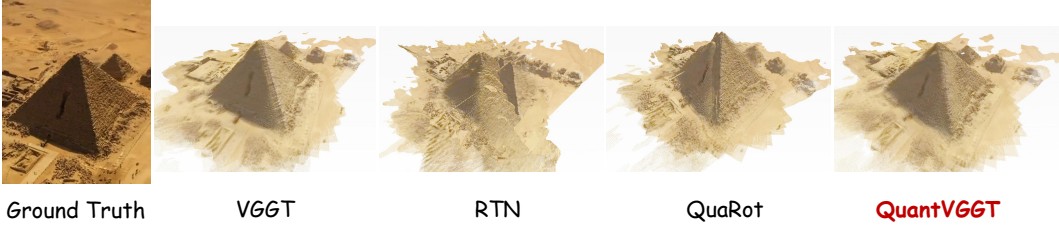

Figure 11: Visual comparison results with more methods.

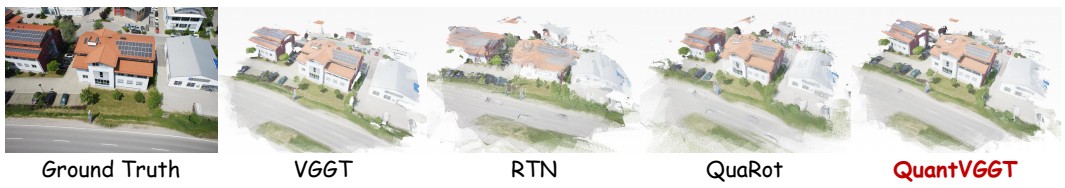

Figure 12: Visual comparison results with more methods.

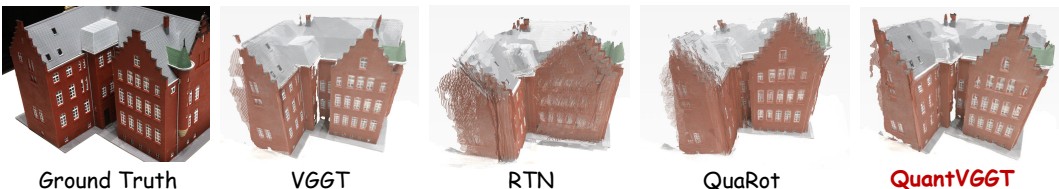

Figure 13: Visual comparison results with more methods.

