# OpenReview forum: "Quantized Visual Geometry Grounded Transformer"
_ICLR.cc/2026/Conference — ICLR 2026 Poster_

### Official Review · Reviewer_9UAK · 2025-10-27

**Soundness:** 3
**Presentation:** 3
**Contribution:** 3
**Rating:** 6
**Confidence:** 2

**Summary:**

This paper proposed QuantVGGT, the first comprehensive post-training quantization framework designed for visual geometry grounded transformer. QuantVGGT consists of two components: Dual-Smoothed Fine-Grained Quantization architecture for smoothing the distribution for better quantization performance from both global and local perspective and Noise-Filtered Diverse Sampling strategy for constructing information-maximized calibration dataset. Extensive experiments on multiple tasks compared with different strong baseline methods shown that QuantVGGT greatly outperforms the compared methods and achieves SOTA performance across different bit-width and tasks.

**Strengths:**

1. The motivation of QuantVGGT is clear and quite meaningful, as a systematic quantization research on large-scale 3D foundational models in is heavily unexplored and crucial.

2. The proposed method is supported by extensive empirical analysis and theoretical theorems, and appears to effectively solve the corresponding problems.

3. The paper writing is very clear and accompanied by key illustration figures and method flowcharts, which are easy to understand and implement.

4. The paper conducted high bit (8-bit) and low bit (4-bit) experiments on multiple task datasets, significantly surpassing strong baseline methods from different fields, demonstrating the effectiveness and generalization of QuantVGGT.

5. QuantVGGT reported the true efficiency in hardware, calibration consumption, various detailed ablation experiments, and extensive experimental analysis. This fully demonstrates the effectiveness of its method and the effectiveness of acceleration in real-world scenarios.

**Weaknesses:**

1. The symbol writing on line 192 seems to be incorrect. And for the compared method in line 377, I believe it should be spelled QuaRot.

2. Providing acceleration effects at different sequence lengths will further enhance its practicality in different scenarios.

3. Adding different comparison methods reconstruction visualization will better demonstrate the visual effectiveness of the proposed methods.

**Questions:**

1. The symbol writing on line 192 seems to be incorrect. And for the compared method in line 377, I believe it should be spelled QuaRot.

2. Providing acceleration effects at different sequence lengths will further enhance its practicality in different scenarios.

3. Adding different comparison methods reconstruction visualization will better demonstrate the visual effectiveness of the proposed methods.

---

> ### Author Response · Authors · 2025-11-22
>
> We appreciate your thorough and constructive feedback. Below we provide detailed responses to each of your comments.
>
> >**Q1:** Writing revision.
>
> Thank you for pointing out the issues in line 192 and line 377. We have corrected the symbol notation in line 192 and fixed the spelling of “QuaRot” in line 377 to ensure accuracy and consistency.
>
> >**Q2:** Acceleration effects at different sequence lengths.
>
> Thank you for this valuable suggestion.
> To better quantify the practical efficiency of our framework under varying computational demands, we conducted an additional experiment evaluating the **inference acceleration of QuantVGGT (W4A4) across different sequence lengths**:
>
> |Length|10|20|40|80|
> |-|-|-|-|-|
> |Latency Optimization$\uparrow$|2.47$\times$|2.49$\times$|2.53$\times$|2.55$\times$|
>
> These results show that even as sequence length increases, QuantVGGT **consistently provides substantial acceleration with stable scaling trends.** We have incorporated the complete results and analysis in the revised manuscript (Appendix Section F Table 15).
>
> >**Q3:** More visual comparison.
>
> Thank you for this insightful suggestion.
> We agree that including additional reconstruction visualizations can further demonstrate the effectiveness of our quantized model.
>
> In the revised manuscript, we have added **more qualitative 3D reconstruction comparisons between QuantVGGT, FP16 VGGT, as well as other baselines**, including QuaRot and RTN in Appendix Section H.
>
> These visualizations clearly show that QuantVGGT preserves fine geometric details and depth consistency, even under aggressive W4A4 quantization. We believe this enhancement strengthens the empirical evidence of our method’s visual fidelity.

---

### Official Review · Reviewer_7NMK · 2025-10-30

**Soundness:** 4
**Presentation:** 3
**Contribution:** 3
**Rating:** 8
**Confidence:** 4

**Summary:**

This submission presents QuantVGGT, the first Post-Training Quantization (PTQ) framework for Visual Geometry Grounded Transformer (VGGT), a state-of-the-art model for learning-based 3D reconstruction. The work addresses two critical challenges in VGGT quantization: (1) heavy-tailed activation distributions induced by data-independent camera and register tokens, and (2) unrepresentative calibration dataset building due to the multi-view complexity of 3D data.
For challenge 1, the authors propose Dual-Smoothed Fine-Grained Quantization (DSFQ), which consists of a pre-global Hadamard rotation for dispersing outliers and smoothing heavy tails, a post-local channel scaling for normalizing inter-channel variance. Furthermore, the authors adopt token-wise quantization for activation and outer-dimension-wise quantization for weight, forming fine-grained quantization, which leads to minor quantization error.
For challenge 2, the authors propose Noise-Filtered Diverse Sampling (NFDS), which filters noisy outliers using deep-layer activation statistics and constructs frame-aware calibration clusters concerning VGGT’s inductive bias of relationships between first-frame and subsequent-frames.
Comprehensive experiments on Co3Dv2 for camera pose estimation and DTU for point map estimation demonstrate that, QuantVGGT outperforms existing quantization methods across various bit-widths both in theoretical validation and real-hardware deployment.

**Strengths:**

1. Originality: Target model quantization on VGGT, a new SOTA 3D reconstruction model, analyzing concrete experimental phenomena, heavy-tailed activation distribution and unrepresentative calibration dataset, and proposing corresponding observation-driven solutions, DSFQ and NFDS.

2. Validation: Detailed mathematical theoretical derivations and justifications for technical innovations. Comprehensive experiments with additional ablation studies across tasks, quantization bit-widths, comparison methods, and proposed components.

3. Clarity: Hierarchical narrative logic, clear writing, well-organized sections, and informative visualizations make complex concepts (e.g., Hadamard rotation and frame-aware sampling) accessible to readers from both 3D vision and quantization backgrounds.

4. Significance: Bridge the gap between large-scale 3D reconstruction model performance and deployment efficiency, with results that are both scientifically influential and practically useful, which advances model quantization in 3D field and enables edge 3D reconstruction.

**Weaknesses:**

1. Insufficient Related Works: Not involve other works on VGGT acceleration and optimization, such as FastVGGT using token compression and FasterVGGT using sparse attention, let alone comparison with these methods in Sec 4 to highlight the efficiency of QuantVGGT.

2. Rough Analysis of Special Tokens: The paper mentions that data-independent special tokens cause heavy tails, but does not explicitly and concretely state the respective influences of camera tokens and register tokens.

3. Experiment Generalization:
(1) Lack of experiments on the widely adopted W4A8 quantization configuration to enable more comprehensive comparisons with existing methods.
(2) Lack of experiments on more tasks (e.g., Multi-view Depth Estimation on DTU, Image Matching on ScanNet-1500) and datasets (e.g., Camera Pose Estimation on RealEstate10K, Point Map Estimation on ETH3D) compared with the original VGGT paper.

4. Experiment Credibility:
(1) In Sec 3, the authors claim that VGGT has an inductive bias for modeling relationships between the first frame and subsequent frames. From the theoretical perspective, this phenomenon perhaps originates from the unique design of VGGT’s two distinct sets of special tokens. However, the paper barely supply sufficient experimental evidence to reflect it.
(2) Activation Distribution only in two adjacent blocks (frame/global block 7/8) in Appendix D fail to confirm the claimed ubiquity of the heavy-tailed phenomenon across different layers.
(3) Lack of more subjective visualization results in Appendix H in across more scenarios, including comparisons between the FP16 baseline, QuantVGGT, and other quantization methods under various quantization settings.

**Questions:**

1. For frame-aware clustering in NFDS, the correlation vector c^i measures similarity between the first frame and subsequent frames. How does this strategy perform on real-world 3D sequences where the first frame is an outlier (e.g., occluded or low-light)?

2. Why does the QuantVGGT in W8A8 outperform the FP16 baseline in Camera Pose Estimation on Co3Dv2? Please analyze the phenomenon and clarify the potential causes. Moreover, provide a reasonable explanation for the marginal performance difference between QuantVGGT and QuaRot under W8A8 setting.

3. Is there a possibility of deploying QAT or extremely low-bit PTQ in VGGT? How should the quantization strategies be adapted or specially designed for these scenarios respectively?

4. Have you ever studied whether other 3D reconstruction models (e.g., DUSt3R, MASt3R) also have similar phenomenon of heavy-tailed activation distribution and unstable calibration? Would DSFQ and NFDS perform well on them?

5. Presentation and Visualization Suggestions:
(1) Supplement brief explanations of variables R_j^*, V_j^*, and V^* directly in Theorem 3.2, and add a hyperlink to Appendix A immediately for quick reference to related details.
(2) Consider enlarging the size of Fig. 4 (a) and supplementing the specific attributes (labels or features) corresponding to each cluster in Fig. 4 (b)(c)(d) in Appendix E to improve readability.
(3) Include a concise introduction to "global robust moments" before Eq. 9.
(4) Give an explicit definition of "static quantization" and "dynamic quantization" and clarify whether "tensor-wise" and "token-wise" refer to activation quantization specifically.
(5) Merge Table 6 in Appendix D into Table 3 of Sec 4.3 to avoid redundant presentation.
(6) Provide detailed hardware platform information together with absolute latency and optimize the presentation format of Fig. 6.
(7) Consider consolidating foundational knowledge of rotation-based quantization (e.g., Hadamard transform) and migration-factor-based quantization (e.g., SmoothQuant) into Sec 3.1 (Preliminary) for better logical coherence.
(8) Add optional ablation studies for hyperparameters, including the filtering threshold T (or p), the cluster number K and the calibration samples scale N.

---

> ### Author Response · Authors · 2025-11-22
>
> We sincerely thank you for your positive assessment of our work and your constructive suggestions. We address your comments in detail below.
>
> >**W1:** Comparison with FastVGGT.
>
> FastVGGT [1] accelerates VGGT via **token compression / token merging**, while our QuantVGGT focuses on **post-training quantization**. These techniques are **conceptually orthogonal** and, in principle, can be combined.
>
> To provide a concrete comparison, we evaluate runtime FLOPs and reconstruction accuracy on Co3Dv2:
>
> | Method        | FLOPs (T)$\downarrow$ | AUC@30$\uparrow$ | AUC@15$\uparrow$ | AUC@5$\uparrow$ | AUC@3$\uparrow$ |
> | ------------- | --------------------- | ---------------- | ---------------- | --------------- | --------------- |
> | Full Prec.    | 5.84                  | 89.5             | 83.2             | 66.1            | 54.9            |
> | FastVGGT      | 1.68                  | 82.7             | 71.3             | 39.2            | 25.1            |
> | **QuantVGGT** | **1.40**              | **86.9**         | **78.7**         | **57.3**        | **43.6**        |
>
> From these results, we observe that:
> * **QuantVGGT achieves larger FLOPs reduction** than FastVGGT (1.40T vs. 1.68T),
> * while also **maintaining significantly higher accuracy** at all metrics.
>
> This highlights that **quantization-based acceleration (QuantVGGT) can be highly competitive** with token-compression-based acceleration (FastVGGT), even though they target different aspects of the model.
>
> [1]. Shen Y, et al. Fastvggt: Training-free acceleration of visual geometry transformer. Arxiv 2025.
>
> >**W2:** Analysis of special tokens.
>
> We appreciate your suggestion. We have clarified and strengthened this analysis in the revision.
>
> As shown in Eq. (2), VGGT concatenates **camera tokens, register tokens**, and visual tokens along the token dimension before feeding them into the transformer blocks.
>
> We have **indexed special tokens separately and visualized** their behaviors in the token heatmaps (Fig. 3(b) in the main text and Fig. 7(e–h) in Appendix E). This shows that **both camera tokens and register tokens exhibit substantially heavier tails and stronger saliency** than regular visual tokens, which explains why they dominate the global activation range and become primary sources of outliers for quantization.
>
> We now state these points explicitly in Sec. 3.2 and Appendix E, making it clear how camera tokens and register tokens respectively contribute to the heavy-tailed activation distributions that motivate our DSFQ design.
>
> >**W3:** Experiments on W4A8 setting, more tasks and datasets.
>
> **Experiment under W4A8**
> We conduct an additional experiment under W4A8 on Co3Dv2 (10 frames), using the same calibration protocol. We use the first 200 sequences of each class for faster evaluation:
>
> |Method|AUC@30$\uparrow$|AUC@15$\uparrow$|AUC@5$\uparrow$|AUC@3$\uparrow$|
> |-|-|-|-|-|
> |SmoothQuant|88.5|81.4|62.1|49.3|
> |QuarRot|89.3|82.6|64.8|53.2|
> |**QuantVGGT**|90.1|83.7|66.0|54.3|
>
> QuantVGGT **consistently outperforms** both SmoothQuant and QuaRot across all metrics. Together with our W8A8 and W4A4 results, this shows that QuantVGGT is effective and robust across different bit-widths, from moderate compression (W4A8) to more aggressive low-bit settings (W4A4).
>
> **Experiment on more datasets**
> To further demonstrate the generalization ability of QuantVGGT beyond Co3Dv2 and DTU, we conduct additional **point cloud reconstruction experiments** on the **7-Scenes** [1] and **NRGBD** [2] datasets under the same W4A4 quantization protocol. The results are summarized below:
>
> |dataset|methord|Acc. Mean$\downarrow$|Acc Med.$\downarrow$|Comp. Mean$\downarrow$|Comp. Med.$\downarrow$|N.C. Mean$\uparrow$|N.C. Med.$\uparrow$|
> |-|-|-|-|-|-|-|-|
> |7-Scenes|Full Prec.|0.025|0.013|0.036|0.020|0.728|0.836|
> ||SmoothQuant|0.370|0.261|0.498|0.361|0.484|0.477|
> ||QuaRot|0.030|0.016|0.042|0.022|0.701|0.800|
> ||QuantVGGT|**0.026**|**0.013**|**0.037**|**0.019**|**0.718**|**0.812**|
> |NRGBD|Full Prec.|0.015|0.009|0.017| 0.007|0.878|0.969|
> ||SmoothQuant|0.479|0.393|0.614|0.489|0.515|0.513|
> ||QuaRot|0.034|0.021|0.030|0.015|0.820|0.948|
> ||QuantVGGT|**0.019**|**0.013**|**0.021**|**0.010**|**0.850**|**0.959**|
>
> On both datasets, QuantVGGT **substantially outperforms** current SOTA methods SmoothQuant and QuaRot, also maintains nearly lossless performance. These results demonstrate that **QuantVGGT retains strong reconstruction quality across diverse benchmarks**, highlighting its robustness and generality.
>
> [1]. Shotton J, et al. Scene coordinate regression forests for camera relocalization in RGB-D images. CVPR 2013.
>
> [2]. Azinović D, et al. Neural rgb-d surface reconstruction. CVPR 2022.

---

> ### Author Response · Authors · 2025-11-22
>
> >**W4:** Experiment Credibility.
>
> 1. **Inductive bias between the first frame and subsequent frames.**
> In Sec. 3, we claimed that VGGT has an inductive bias to model relationships between the first frame and subsequent frames, partly due to its two sets of special tokens. To provide more concrete evidence, we add a new experiment:
>     * Given a 10-frame input sequence, we **systematically change which frame is treated as the “first” frame**, while keeping all other settings fixed, and re-run camera pose estimation.
>     * We then measure the AUC@30 under different choices of the first frame.
>
> |First Frame index|0|1|2|3|4|5|6|7|
> |-|-|-|-|-|-|-|-|-|
> |AUC@30|86.9|86.2|86.5|87.2|87.0|86.4|86.1|86.7|
>
> Empirically, we observe a variance in pose accuracy when the first frame is changed. This confirms that VGGT indeed builds an asymmetric dependency structure anchored at the first frame, and that the special token design indeed induces a learned preference for first–subsequent frame relations.
>
> 2. **Heavy-tailed activations across layers.**
>     * **Multi-layer outlier visualization.**
>     Figure 3(a) and Figure 7(a)(b)\(c)(d) are **five representative layers** across the network, including both frame-attention and global-attention blocks. These plots show that: **heavy-tailed outliers appear consistently across different attention layers**, with varying magnitude but the same characteristic shape (few very large activations dominating the numerical range).
>     * **Special-token saliency across layers.**
>     Beyond raw activation distributions, we additionally visualize **saliency maps of special tokens across five layers**. The visualizations (shown in Figure 3(b) and Figure 7(e)(f)(g)(h)) consistently indicate that **special tokens carry much larger activations and saliency** than regular visual tokens in multiple layers, confirming that they are persistent sources of heavy-tailed behavior.
> 3. **Additional subjective visualization across methods and settings.**
> In the revised manuscript, we have added **more qualitative 3D reconstruction comparisons between QuantVGGT, FP16 VGGT, as well as other baselines**, including QuaRot and RTN in xx.
> These visualizations clearly show that QuantVGGT preserves fine geometric details and depth consistency, even under aggressive W4A4 quantization. We believe this enhancement strengthens the empirical evidence of our method’s visual fidelity.

---

> ### Author Response · Authors · 2025-11-22
>
> >**Q1:** NFDS performance when the first frame is an outlier.
>
> To evaluate NFDS under extreme cases, we construct a subset from CO3Dv2 dataset where the *first frame is significantly inconsistent* with the rest of the sequence. Specifically, we filter calibration candidates whose average cosine similarity between the first frame and all other frames is **< 0.1**. On this challenging subset, we compare Random sampling and NFDS:
>
> |Method|AUC@30$\uparrow$|AUC@15$\uparrow$|AUC@5$\uparrow$|AUC@3$\uparrow$|
> |-|-|-|-|-|
> |Full Prec.|90.5|84.4|67.9|57.0|
> |Random|87.7|78.6|57.6|43.4|
> |**NFDS**|**87.9**|**79.0**|**58.4**|**44.7**|
>
> NFDS consistently outperforms random sampling because:
>
> 1. **Outlier filtering** removes extreme low-quality samples before clustering.
> 2. **Frame-aware clustering is not solely dependent on the first frame**, since clustering focuses on relative inter-frame geometry, rather than absolute appearance quality.
>
> We added this discussion and experiment in the revised Appendix xx.
>
> >**Q2:** Explanation of QuantVGGT's performance under W8A8.
>
> We appreciate the reviewer for pointing out this phenomenon. We provide our analysis and clarification below.
>
> **Why does W8A8 outperform FP16 baseline on certain metrics?**
> We appreciate the reviewer’s careful observation. We view this as a small but reasonable effect of our calibration strategy rather than a fundamental advantage of low-bit over full precision. First, NFDS focuses calibration on **informative and diverse 3D sequences**, so the quantization ranges are better aligned with the “important” regions of the data distribution than the original FP16 model. Second, DSFQ and W8A8 quantization **suppress extreme outliers and noisy activations**, which acts as a mild regularizer and can slightly improve robustness on some metrics.
>
> **Why is the performance gap between QuantVGGT and QuaRot under W8A8 only marginal?**
> we note that **W8A8 is a relatively high-precision setting**, where the quantization error is inherently small. A strong rotation-based baseline such as QuaRot already removes much of the quantization-induced distortion at 8 bits, so the remaining headroom for further improvement is necessarily limited. In this regime:
> * QuaRot already produces an almost lossless W8A8 model (e.g., 54.6 vs. 54.9 AUC@3 for FP),
> * QuantVGGT still provides **consistent improvements across all metrics**, but the absolute gap is modest because **both methods are close to the FP ceiling**.
>
> In contrast, for more aggressive settings such as **W4A4**, the quantization error is much larger and the advantages of DSFQ (outlier suppression) and NFDS (3D-aware calibration) become more pronounced. As reported in Table 1 and 2, under W4A4 our QuantVGGT achieves **86.9 AUC@30**, significantly outperforming QuaRot’s **81.8** and yielding large gains across all thresholds. This indicates that **QuantVGGT brings consistent performance improvements at both high and low bit-widths**, with especially strong benefits in the more challenging low-bit regime where quantization errors dominate.

---

> ### Author Response · Authors · 2025-11-22
>
> >**Q3:** Applying Quantization-Aware Training (QAT) to QuantVGGT.
>
> Thank you for the valuable suggestion. Here we provide a detailed discussion:
>
> **Feasibility and cost of QAT on VGGT-1B.**
> In principle, QuantVGGT can be combined with QAT. However, applying full QAT to VGGT-1B is extremely resource-intensive:
> * VGGT-1B [1] is trained on large-scale multi-view data (millions of images from more than 17 datasets) with heavy computation.
> * The original training of VGGT-1B requires **> 500 A100 GPU-days**, which is well beyond the typical deployment budget for PTQ scenarios.
>
> **A small-scale QAT study on QuantVGGT.**
> Due to the limited computation resources but to still provide empirical evidence, we conduct a lightweight QAT experiment by fine-tuning the quantized model for **1 epoch** on the **same calibration dataset** used by QuantVGGT (i.e., no extra data are introduced). The results are:
>
> |Method|GPU Memory (GB)|GPU Time (Hours)|AUC@30$\uparrow$|AUC@15$\uparrow$|AUC@5$\uparrow$|AUC@3$\uparrow$|
> |-|-|-|-|-|-|-|
> |Full Prec.|-|-|90.5|84.4|67.9|57.0|
> |**QuantVGGT (PTQ)**|14.6|2.67|87.9|79.0|58.4|44.7|
> |**QuantVGGT (QAT)**|53.7|9.50|88.1|80.2|58.6|44.6|
>
> We draw the following observations:
> * **QAT indeed improves low-bit performance.** Even with only 1 epoch on a small calibration subset, QAT brings consistent gains over pure PTQ (e.g., +1.2 AUC@15, +0.2 AUC@30), confirming that **applying QAT to QuantVGGT is effective and has clear potential** to further close the gap to the full-precision model.
> * **The computational overhead is substantial.** Achieving these modest gains requires **~3.7× GPU memory** (14.6 → 53.7 GB) and **>3.5× GPU time** (2.67 → 9.50 hours) even in this very limited setting, whereas our main results focus on efficient PTQ with minimal resource requirements.
>
> **Positioning relative to our work.**
> Our paper is intentionally centered on **efficient PTQ** that can be deployed without revisiting large-scale training. The above experiment shows that:
> * QAT on top of QuantVGGT is **meaningful and beneficial** for low-bit quantization;
> * It is also **computationally much more demanding**, making it a different regime than the deployment-oriented PTQ setting we target.
>
> We will add this discussion and the table to xx of the revised manuscript, and explicitly clarify that **QAT is complementary rather than conflicting** with our method: QuantVGGT provides a strong PTQ baseline that can serve either as a standalone solution or as **a good initialization for subsequent QAT**, depending on the available computational budget.

---

> ### Author Response · Authors · 2025-11-22
>
> >**Q4:** Experiments on other 3D backbones.
>
> To validate QuantVGGT generality across architectures, we additionally evaluate QuantVGGT on **Fast3R** [1], a recently proposed transformer-based 3D reconstruction backbone. We report point-cloud reconstruction results on 7-Scenes [2] dataset:
>
> |Method|Acc. Mean$\downarrow$|Acc Med.$\downarrow$|Comp. Mean$\downarrow$|Comp. Med.$\downarrow$|N.C. Mean$\uparrow$|N.C. Med.$\uparrow$|
> |-|-|-|-|-|-|-|
> |Full Prec.|0.049|0.021|0.069|0.021|0.624|0.683|
> |SmoothQuant|0.497|0.448|0.319|0.244|0.586|0.625|
> |QuaRot|0.312|0.258|0.149|0.080|0.593|0.637|
> |**QuantVGGT**|**0.049**|**0.022**|**0.070**|**0.022**|**0.624**|**0.683**|
>
> QuantVGGT achieves **practically lossless performance** relative to the full-precision Fast3R model and **substantially surpasses** strong quantization baselines (QuaRot and SmoothQuant) on all metrics.
>
> These results highlight the following points:
> 1. **DSFQ targets a universal outlier problem, while being especially well aligned with VGGT.**
> The design of DSFQ is directly motivated by the severe heavy-tailed outliers observed in VGGT, which are particularly amplified by VGGT’s special tokens. In this sense, DSFQ is naturally tailored to VGGT. At the same time, its mechanism depends only on empirical activation statistics: it rescales and regularizes heavy-tailed outliers regardless of where the outliers arise from. The Fast3R results show that our DSFQ achieves practically lossless performance and clearly surpasses SmoothQuant and QuaRot, demonstrating that this VGGT-motivated design in fact resolves a universal outlier issue shared by many large 3D transformers.
> 2. **NFDS exploits intrinsic 3D structure, while being naturally matched to VGGT’s inductive bias.**
> NFDS is derived from our sampling principle (Theorem 3.2) and only assumes that the inputs are **multi-view 3D sequences**. It uses frame-wise correlation metrics for diverse sampling and depends solely on the natural 3D geometry and temporal structure of the data. VGGT’s special tokens and first–subsequent frame design amplify these frame-wise correlations, making NFDS particularly effective and well aligned with VGGT’s inductive bias. Importantly, when we apply QuantVGGT to Fast3R, we still observe **lossless performance**, whereas generic PTQ baselines degrade substantially. This confirms that NFDS is **not restricted to VGGT**, but benefits any 3D backbone that exhibits multi-view correlations.
> 3. **Overall message.**
> The lossless performance on Fast3R shows that our **VGGT-tailored designs remain architecture-agnostic at the principle level**:
>     * They are inspired by and particularly suitable for VGGT due to its strong inductive bias and pronounced outlier behavior;
>     * Yet they **address fundamental, model-agnostic PTQ challenges**: universal activation outliers (DSFQ) and 3D correlation-aware calibration (NFDS). And therefore can be directly transferred to other 3D backbones.
>
> [1]. Yang J, et al. Fast3r: Towards 3d reconstruction of 1000+ images in one forward pass. CVPR 2025.
>
> [2]. Shotton J, et al. Scene coordinate regression forests for camera relocalization in RGB-D images. CVPR 2013.
>
> >**Q5:** Presentation and visualization revision.
>
> We sincerely appreciate these suggestions. We have incorporated all of them:
>
> 1. **Added explicit definitions** of $𝑅_𝑗^*$, $V_𝑗^*$, $V^*$ in Theorem 3.2.
> 2. **Added hyperlink** from Sec. 3.3 to Appendix A.
> 3. **Introduced global robust moments** before Eq. (9).
> 4. Clarified definitions of **static vs. dynamic quantization**, and of tensor-wise vs. token-wise activation quantization in Table 5 and Appendix Section D.
> 5. **Merged Table 6 into Table 3** for conciseness.
> 6. **Added hardware details** and absolute latency values.
> 7. Reorganized Sec. 3.1 to include background on Hadamard rotation and SmoothQuant.
> 8. Added ablations for **threshold p, cluster number K**, and **calibration sample size N**. The results (W4A4 QuantVGGT on Co3Dv2, 10 frames) are summarized as follows. We use the first 200 sequences of each class for faster evaluation:
>
> |Method|AUC@30$\uparrow$|AUC@15$\uparrow$|AUC@5$\uparrow$|AUC@3$\uparrow$|
> |-|-|-|-|-|
> |Full Prec.|90.5|84.4|67.9|57.0|
> |QuaRot|82.8|71.4|41.0|24.4|
> |**QuantVGGT ($p=0.1$)**|87.5|78.7|57.8|44.2|
> |**QuantVGGT ($p=0.2$)**|87.9|79.0|58.4|44.7|
> |**QuantVGGT ($p=0.3$)**|87.5|78.8|57.8|44.3|
> |**QuantVGGT ($K=6$)**|87.7|79.7|58.1|44.4|
> |**QuantVGGT ($K=8$)**|87.9|79.0|58.4|44.7|
> |**QuantVGGT ($K=10$)**|87.7|79.7|57.9|44.0|
> |**QuantVGGT (5 samples)**|86.4|77.6|53.8|38.9|
> |**QuantVGGT (10 samples)**|86.9|78.4|55.3|40.8|
> |**QuantVGGT (20 samples)**|86.9|78.4|55.3|40.8|
> |**QuantVGGT (40 samples)**|87.9|79.0|58.4|44.7|
>
> These results show that QuantVGGT is **robust over a wide range of hyperparameters**, and performance steadily improves with more calibration samples.

---

### Official Review · Reviewer_o5ar · 2025-10-31

**Soundness:** 3
**Presentation:** 3
**Contribution:** 3
**Rating:** 6
**Confidence:** 3

**Summary:**

This paper introduces QuantVGGT, the first post-training quantization framework specifically designed for billion-parameter Visual Geometry Grounded Transformers (VGGTs), tackling two unique challenges in 3D reconstruction: (1) Dual-Smoothed Fine-Grained Quantization (DSFQ) employs a pre-global Hadamard rotation to disperse heavy-tailed activations from data-independent tokens, followed by post-local channel smoothing to stabilize variance without runtime overhead; and (2) Noise-Filtered Diverse Sampling (NFDS) filters outliers using deep-layer statistics and constructs calibration clusters based on VGGT’s intrinsic frame-relative geometric bias, ensuring representative sampling. The approach achieves state-of-the-art results under 4-bit quantization, preserving over 98% of full-precision accuracy while delivering 2.5× speedup and 3.7× memory compression—enabling practical deployment of large 3D vision models in resource-constrained environments.

**Strengths:**

S1: Novel and Insightful Problem Formulation. The paper's core originality lies not in inventing a new algorithm, but in being the first to identify and formalize the unique quantization challenges specific to 3D vision transformers—namely, the instability caused by data-independent special tokens and multi-view calibration. This re-framing of the problem is a significant contribution in itself.

S2: High-Quality and Tailored Technical Solutions. The proposed methods (DSFQ and NFDS) are technically robust and specifically tailored to the identified problems. In particular, NFDS cleverly leverages the model's own geometric inductive bias for sampling, demonstrating a deep understanding of the architecture that goes beyond generic quantization techniques.

S3: High Practical Significance and Impact. The work's significance is substantial, as it bridges the gap between powerful, large-scale 3D models and real-world deployment. Achieving 98% accuracy at 4-bit with significant memory and latency reduction makes these models viable for edge devices, unlocking new possibilities in robotics, AR/VR, and other resource-constrained applications.

S4: Excellent Clarity and Presentation. The paper is exceptionally clear and well-written. The use of informative figures to visualize abstract concepts like activation distributions, combined with intuitive and rigorous explanations, makes the complex contributions accessible and highly convincing.

**Weaknesses:**

W1: Lack of Real-World Hardware Benchmarks. The reported latency and memory gains are theoretical. The paper lacks empirical validation on actual hardware using standard inference engines (e.g., TensorRT, ONNX Runtime), making it difficult to confirm if the claimed speedups translate to real-world deployment.

W2: Unclear Sensitivity to Calibration Data Size. The robustness of the method to the number of calibration samples is not explored. An ablation study on performance with significantly fewer samples (e.g., 10 or 20) is missing, which is critical for understanding its practicality in data-scarce scenarios.

W3: Limited Generalization to Other Architectures. The evaluation is limited exclusively to the VGGT architecture. It remains unclear if the proposed solutions are generalizable to other 3D vision transformers (e.g., DUSt3R, MASt3R), which limits the broader impact of the work.

W4: Missing Comparison to Quantization-Aware Training (QAT). The paper does not include a comparison to QAT. A QAT baseline, even with minimal fine-tuning, would provide crucial context on the absolute performance ceiling and help better evaluate the effectiveness of this PTQ-only approach.

**Questions:**

Q1: How does performance degrade as the calibration set size is significantly reduced (e.g., to 20, 10, or 5 samples)?

Q2: In the extreme low-data regime (e.g., <10 samples), does your NFDS method maintain superior stability (lower variance) compared to random sampling?

Q3: How specific are the identified quantization challenges to VGGT? Could the core principles of DSFQ and NFDS be applied to other 3D vision transformers like DUSt3R or MASt3R?

Q4: Could you clarify what "random Hadamard matrix" means? How was it generated, and is the model's performance sensitive to the specific matrix chosen?

Q5: Is the remaining performance gap at W4A4 a fundamental limitation of PTQ for this model? How would it compare against a minimally fine-tuned QAT baseline (e.g., after 1 epoch)?

**Details Of Ethics Concerns:**

/

---

> ### Author Response · Authors · 2025-11-22
>
> We are deeply grateful for your support of our work, and we provide detailed responses to your comments as follows:
>
> >**W1:** Real-world hardware benchmarks.
>
> We are sorry for the misunderstanding. **We clarify that the reported latency and memory gains are collected from real hardware.**
>
> We clarify this by adding "All the experiments are conducted on a single NVIDIA RTX 4090 24G GPU with CUDA 12.4. We use CUTLASS on top of PyTorch for performing low-bit INT matrix multiplication." in the revised manuscript Section 4.4.
>
> >**W2 & Q1 & Q2:** Ablation study on calibration data size.
>
> Thank you for pointing out the importance of calibration data size. We have added an ablation study that explicitly evaluates QuantVGGT with significantly fewer calibration sequences (5, 10, 20, and 40), using the same W4A4 Co3Dv2 camera pose estimation setup:
>
> |Data Size|AUC@30$\uparrow$|AUC@15$\uparrow$|AUC@5$\uparrow$|AUC@3$\uparrow$|
> |-|-|-|-|-|
> |Full Prec.|90.5|84.4|67.9|57.0|
> |QuaRot (40 samples)|82.8|71.4|41.0|24.4|
> |**QuantVGGT (5 samples)**|86.4|77.6|53.8|38.9|
> |**QuantVGGT (10 samples)**|86.5|77.6|53.6|38.5|
> |**QuantVGGT (20 samples)**|86.9|78.4|55.3|40.8|
> |**QuantVGGT (40 samples)**|87.9|79.0|58.4|44.7|
>
> From these results we observe that:
> * **Performance degrades gracefully** as the calibration set size is reduced from 40 → 20 → 10 → 5 samples, with **no abrupt collapse** even at 5 samples.
> * Even in the **extreme low-data setting (5 samples)**, QuantVGGT remains **clearly better than QuaRot** (e.g., 86.4 vs 82.8 AUC@30 and 38.9 vs 24.4 AUC@3), indicating that the proposed DSFQ + NFDS pipeline is effective even with very limited calibration data.
> * With **20 samples**, QuantVGGT already recovers most of the performance of the 40-sample setting (4.1 gain for 20 samples and 5.1 for 40 samples), suggesting good practicality in moderately data-scarce scenarios.
>
> To directly address the question about stability under very small calibration budgets, we further compare **NFDS vs. random sampling** in the 5-sample regime. We report mean±std over 5 runs:
>
> |Method|AUC@30$\uparrow$|
> |-|-|
> |Random (5 samples)|84.9$_{\pm1.2}$|
> |**NFDS (5 samples)**|**86.4$_{\pm0.3}$**|
>
> This shows that:
> * NFDS not only **improves the mean performance** (86.4 vs 84.9 AUC@30),
> * but also achieves **substantially lower variance** (±0.3 vs ±1.2), indicating **higher stability** even under extreme low-data conditions.
>
> This behavior is consistent with the design of NFDS: instead of randomly picking sequences, NFDS explicitly exploits multi-view correlation structure to select **diverse and representative** calibration samples. We will include these ablation results and a brief discussion in Sec. 4.3 / Appendix, and we will emphasize that QuantVGGT is **robust to calibration data size**, and that **NFDS maintains superior stability and accuracy even with very few calibration sequences**.

---

> ### Author Response · Authors · 2025-11-22
>
> >**W3 & Q3:** Experiments on other 3D backbones.
>
> To validate QuantVGGT generality across architectures, we additionally evaluate QuantVGGT on **Fast3R** [1], a recently proposed transformer-based 3D reconstruction backbone. We report point-cloud reconstruction results on 7-Scenes [2] dataset:
>
> |Method|Acc. Mean$\downarrow$|Acc Med.$\downarrow$|Comp. Mean$\downarrow$|Comp. Med.$\downarrow$|N.C. Mean$\uparrow$|N.C. Med.$\uparrow$|
> |-|-|-|-|-|-|-|
> |Full Prec.|0.049|0.021|0.069|0.021|0.624|0.683|
> |SmoothQuant|0.497|0.448|0.319|0.244|0.586|0.625|
> |QuaRot|0.312|0.258|0.149|0.080|0.593|0.637|
> |**QuantVGGT**|**0.049**|**0.022**|**0.070**|**0.022**|**0.624**|**0.683**|
>
> QuantVGGT achieves **practically lossless performance** relative to the full-precision Fast3R model and **substantially surpasses** strong quantization baselines (QuaRot and SmoothQuant) on all metrics.
>
> These results highlight the following points:
> 1. **DSFQ targets a universal outlier problem, while being especially well aligned with VGGT.**
> The design of DSFQ is directly motivated by the severe heavy-tailed outliers observed in VGGT, which are particularly amplified by VGGT’s special tokens. In this sense, DSFQ is naturally tailored to VGGT. At the same time, its mechanism depends only on empirical activation statistics: it rescales and regularizes heavy-tailed outliers regardless of where the outliers arise from. The Fast3R results show that our DSFQ achieves practically lossless performance and clearly surpasses SmoothQuant and QuaRot, demonstrating that this VGGT-motivated design in fact resolves a universal outlier issue shared by many large 3D transformers.
> 2. **NFDS exploits intrinsic 3D structure, while being naturally matched to VGGT’s inductive bias.**
> NFDS is derived from our sampling principle (Theorem 3.2) and only assumes that the inputs are **multi-view 3D sequences**. It uses frame-wise correlation metrics for diverse sampling and depends solely on the natural 3D geometry and temporal structure of the data. VGGT’s special tokens and first–subsequent frame design amplify these frame-wise correlations, making NFDS particularly effective and well aligned with VGGT’s inductive bias. Importantly, when we apply QuantVGGT to Fast3R, we still observe **lossless performance**, whereas generic PTQ baselines degrade substantially. This confirms that NFDS is **not restricted to VGGT**, but benefits any 3D backbone that exhibits multi-view correlations.
> 3. **Overall message.**
> The lossless performance on Fast3R shows that our **VGGT-tailored designs remain architecture-agnostic at the principle level**:
>     * They are inspired by and particularly suitable for VGGT due to its strong inductive bias and pronounced outlier behavior;
>     * Yet they **address fundamental, model-agnostic PTQ challenges**: universal activation outliers (DSFQ) and 3D correlation-aware calibration (NFDS). And therefore can be directly transferred to other 3D backbones.
>
> [1]. Yang J, et al. Fast3r: Towards 3d reconstruction of 1000+ images in one forward pass. CVPR 2025.
>
> [2]. Shotton J, et al. Scene coordinate regression forests for camera relocalization in RGB-D images. CVPR 2013.

---

> ### Author Response · Authors · 2025-11-22
>
> >**W4 & Q5:** Comparison to Qantization-Aware Training (QAT).
>
> Thank you for the valuable suggestion. Here we provide a detailed discussion:
>
> **Feasibility and cost of QAT on VGGT-1B.**
> In principle, QuantVGGT can be combined with QAT. However, applying full QAT to VGGT-1B is extremely resource-intensive:
> * VGGT-1B [1] is trained on large-scale multi-view data (millions of images from more than 17 datasets) with heavy computation.
> * The original training of VGGT-1B requires **> 500 A100 GPU-days**, which is well beyond the typical deployment budget for PTQ scenarios.
>
> **A small-scale QAT study on QuantVGGT.**
> Due to the limited computation resources but to still provide empirical evidence, we conduct a lightweight QAT experiment by fine-tuning the quantized model for **1 epoch** on the **same calibration dataset** used by QuantVGGT (i.e., no extra data are introduced). The results are:
>
> |Method|GPU Memory (GB)|GPU Time (Hours)|AUC@30$\uparrow$|AUC@15$\uparrow$|AUC@5$\uparrow$|AUC@3$\uparrow$|
> |-|-|-|-|-|-|-|
> |Full Prec.|-|-|90.5|84.4|67.9|57.0|
> |**QuantVGGT (PTQ)**|14.6|2.67|87.9|79.0|58.4|44.7|
> |**QuantVGGT (QAT)**|53.7|9.50|88.1|80.2|58.6|44.6|
>
> We draw the following observations:
> * **QAT indeed improves low-bit performance.** Even with only 1 epoch on a small calibration subset, QAT brings consistent gains over pure PTQ (e.g., +1.2 AUC@15, +0.2 AUC@30), confirming that **applying QAT to QuantVGGT is effective and has clear potential** to further close the gap to the full-precision model.
> * **The computational overhead is substantial.** Achieving these modest gains requires **~3.7× GPU memory** (14.6 → 53.7 GB) and **>3.5× GPU time** (2.67 → 9.50 hours) even in this very limited setting, whereas our main results focus on efficient PTQ with minimal resource requirements.
>
> **Positioning relative to our work.**
> Our paper is intentionally centered on **efficient PTQ** that can be deployed without revisiting large-scale training. The above experiment shows that:
> * QAT on top of QuantVGGT is **meaningful and beneficial** for low-bit quantization;
> * It is also **computationally much more demanding**, making it a different regime than the deployment-oriented PTQ setting we target.
>
> We will add this discussion and the table to xx of the revised manuscript, and explicitly clarify that **QAT is complementary rather than conflicting** with our method: QuantVGGT provides a strong PTQ baseline that can serve either as a standalone solution or as **a good initialization for subsequent QAT**, depending on the available computational budget.

---

> ### Author Response · Authors · 2025-11-22
>
> >**Q1:** Performance degrade as the calibration set size is significantly reduced (e.g., to 20, 10, or 5 samples).
>
> Please see the reply on ***W2 & Q1 & Q2***.
>
>
> >**Q2:** In the extreme low-data regime (e.g., <10 samples), does your NFDS method maintain superior stability (lower variance) compared to random sampling?
>
> Please see the reply on ***W2 & Q1 & Q2***.
>
> >**Q3:** Could the core principles of DSFQ and NFDS be applied to other 3D vision transformers?
>
> Please see the reply on ***W3 & Q3***.
>
> >**Q4:** Detailed explanation and ablation study on Hadamard matrix.
>
> We are sorry for the confusion. Here we give a detailed explanation.
>
> Given a target dimension $d=2^n$, we can generate the corresponding Hadamard matrix $\hat{\mathbf{H}} _{2^n}$ through
> $$
> \hat{\mathbf{H}} _2=\frac{1}{\sqrt{2}} \begin{bmatrix}1&1 \\\\ 1&-1\end{bmatrix}，\hat{\mathbf{H}} _{2^n}=\hat{\mathbf{H}} _2 \otimes \hat{\mathbf{H}} _{2^{n-1}}.
> $$
> For dimension that $d \neq 2^n$, we can factorize $d=2^nm$ and have $\hat{\mathbf{H}} _{d}=\hat{\mathbf{H}} _{2^n}\otimes \hat{\mathbf{H}} _{m}$. We denote $v$ as a vector containing random draws from $\{+1,-1\}$, and a random Hadamard matrix $\mathbf{H}=\hat{\mathbf{H}}\text{diag}(v)$. This random Hadamard matrix satisfies $\mathbf{H}^{\top}\mathbf{H}=\mathbf{I}$. In our work, the Hadamard matrix is used to **pre-rotate the distribution for smoothing the outliers** impact on quantization as proved in Lemma 3.1.
>
>
> Overall, the random Hadamard matrix are not determined. Previous work [1][2] indentify that the random Hadamard matrix works well on Large Language Models quantization and we follow their implementation to generate Hadamard matrix. We also further conduct an **ablation study on the sensitivity of Hadamard matrix** selection:
>
> |Method|AUC@30$\uparrow$|AUC@15$\uparrow$|AUC@5$\uparrow$|AUC@3$\uparrow$|
> |-|-|-|-|-|
> |Full Prec.|89.5|83.2|66.1|54.9|
> |QuaRot|81.8|70.3|40.1|23.5|
> |**QuantVGGT (seed 1)**|86.9|78.7|57.3|43.6|
> |**QuantVGGT (seed 2)**|86.8|78.6|57.1|43.5|
> |**QuantVGGT (seed 3)**|86.9|78.7|57.3|43.7|
>
> Changing the Hadamard seed has almost no effect (AUC@30 varies within 0.1). This indicates the **random Hadamard matrix is stable** which proves the effectiveness of our proposed DSFQ.
>
> [1]. Ashkboos, et al. QuaRot: Outlier-Free 4-Bit Inference in Rotated LLMs. NeurIPS 2024.
>
> [2]. Tseng A, et al. Quip#: Even better llm quantization with hadamard incoherence and lattice codebooks. ICML 2024.
>
> >**Q5:** Comparison against a minimally fine-tuned QAT baseline.
>
> Please see the reply on ***W4 & Q5***.

---

### Official Review · Reviewer_48HL · 2025-11-01

**Soundness:** 2
**Presentation:** 3
**Contribution:** 2
**Rating:** 6
**Confidence:** 3

**Summary:**

This paper presents QuantVGGT, the first post-training quantization (PTQ) framework specifically designed for Visual Geometry Grounded Transformers (VGGTs) they propose:
Dual-Smoothed Fine-Grained Quantization (DSFQ) — applying a Hadamard pre-rotation and channel-wise post-smoothing to reduce outlier sensitivity and inter-channel variance.
Noise-Filtered Diverse Sampling (NFDS) — filtering outliers using deep-layer activation statistics and building frame-aware diverse calibration clusters for robust calibration.

**Strengths:**

This is the first quantization framework tailored for billion-scale 3D transformers, a meaningful step for deploying 3D vision foundation models efficiently.

Ablation studies clearly isolate contributions from DSFQ and NFDS.

Quantizing VGGT can significantly lower compute/memory costs for 3D perception and reconstruction in real-world systems

**Weaknesses:**

Only two datasets (Co3Dv2, DTU) are used. Broader testing on outdoor scenes could strengthen claims of generality.

The DSFQ pipeline introduces additional preprocessing (Hadamard transform, smoothing, and clustering). The cost and integration details for deployment (e.g., on mobile devices) are not deeply analyzed.

The paper only focuses on PTQ, so it’s unclear how much performance could be recovered with small-scale fine-tuning.

Theorem 3.2 is theoretically elegant but its real contribution to calibration robustness could be better demonstrated through comparative sampling visualizations or empirical sensitivity analysis.

**Questions:**

Do the authors plan to further validate the robustness of their method on more diverse 3D datasets (e.g., ScanNet, DL3DV, or Tanks and Temples)?


How does QuantVGGT perform under mixed precision or asymmetric quantization setups?

How sensitive is QuantVGGT to the choice of Hadamard matrix and smoothing coefficient α?
Would small perturbations in these hyperparameters impact stability?

Does the method generalize to other architectures such as DUSt3R or diffusion-based 3D transformers?

---

> ### Author Response · Authors · 2025-11-22
>
> We sincerely thank the reviewer for the thoughtful and constructive feedback. Below we provide detailed responses to each of your comments.
>
> >**W1 & Q1:** Experiments on 3D more tasks and datasets.
>
> To further demonstrate the generalization ability of QuantVGGT beyond Co3Dv2 and DTU, we conduct additional **point cloud reconstruction experiments** on the **7-Scenes** [1] and **NRGBD** [2] datasets under the same W4A4 quantization protocol. The results are summarized below:
>
> |dataset|methord|Acc. Mean$\downarrow$|Acc Med.$\downarrow$|Comp. Mean$\downarrow$|Comp. Med.$\downarrow$|N.C. Mean$\uparrow$|N.C. Med.$\uparrow$|
> |-|-|-|-|-|-|-|-|
> |7-Scenes|Full Prec.|0.025|0.013|0.036|0.020|0.728|0.836|
> ||SmoothQuant|0.370|0.261|0.498|0.361|0.484|0.477|
> ||QuaRot|0.030|0.016|0.042|0.022|0.701|0.800|
> ||QuantVGGT|**0.026**|**0.013**|**0.037**|**0.019**|**0.718**|**0.812**|
> |NRGBD|Full Prec.|0.015|0.009|0.017| 0.007|0.878|0.969|
> ||SmoothQuant|0.479|0.393|0.614|0.489|0.515|0.513|
> ||QuaRot|0.034|0.021|0.030|0.015|0.820|0.948|
> ||QuantVGGT|**0.019**|**0.013**|**0.021**|**0.010**|**0.850**|**0.959**|
>
> On both datasets, QuantVGGT **substantially outperforms** current SOTA methods SmoothQuant and QuaRot, also maintains nearly lossless performance. These results demonstrate that **QuantVGGT retains strong reconstruction quality across diverse benchmarks**, highlighting its robustness and generality.
>
> [1]. Shotton J, et al. Scene coordinate regression forests for camera relocalization in RGB-D images. CVPR 2013.
>
> [2]. Azinović D, et al. Neural rgb-d surface reconstruction. CVPR 2022.
>
> >**W2:** Additional preprocessing costs and deployment details.
>
> We appreciate the request for clearer deployment analysis. Below we summarize the overhead introduced by each component in the QuantVGGT pipeline.
>
> 1. **Clustering Cost in NFDS.**
> We provide the detailed clustering overhead from Appendix Section F and reproduce the quantitative comparison here:
>
> |Method|GPU Memory$\downarrow$|GPU Time$\downarrow$|AUC@30$\uparrow$|AUC@5$\uparrow$|
> |-|-|-|-|-|
> |Full Prec.|-|-|90.0|83.9|
> |Naive PTQ|14.4|2.53|78.8|65.2|
> |**QuantVGGT (w/o filter)**|14.6$_{+0.2}$|2.56$_{+0.03}$|86.2$_{+7.4}$|77.1$_{+11.9}$|
> |**QuantVGGT (w/o cluster)**|14.6$_{+0.2}$|2.64$_{+0.11}$|86.2$_{+7.4}$|76.5$_{+11.3}$|
> |**QuantVGGT**|14.6$_{+0.2}$|2.67$_{+0.14}$|86.9$_{+9.6}$|78.7$_{+14.9}$|
>
> The clustering step requires only **a single forward pass per sample**, introducing **negligible additional overhead** while yielding **+9.6 improvement** on AUC@30 over naive PTQ.
>
> 2. **Transform Cost in DSFQ.**
>     * **Hadamard Transform**
> We adopt the CUDA-optimized fast-hadamard-transform following QuaRot [1]. This operation is lightweight and adds **almost no runtime overhead**.
>     * Smoothing Factor Fusion
>     Following SmoothQuant [2], the smoothing factor is fused into the preceding LayerNorm during inference, adding **zero runtime cost**.
>
> The inference overhead under W4A4 quantization is summarized below:
>
> |Method|Memory Optimization$\uparrow$|Latency Optimization$\uparrow$|AUC@30$\uparrow$|
> |-|-|-|-|
> |QuantVGGT (w/o Hadamard)|3.65$\times$|2.50$\times$|81.9|
> |QuantVGGT|3.65$\times$|2.49$\times$|**86.9**|
>
> With fast-hadamard-transform CUDA kernel, the Hadamard transform introduces **almost no extra burden** but **significantly improves model performance** from 81.9 AUC@30 to 86.9. This proves both the efficiency and effectiveness of the proposed DSFQ pipeline.

---

> ### Author Response · Authors · 2025-11-22
>
> >**W3:** QuantVGGT performance under small-scale fine-tuning.
>
> Thank you for the valuable suggestion. Here we provide a detailed discussion:
>
> **Feasibility and cost of QAT on VGGT-1B.**
> In principle, QuantVGGT can be combined with QAT. However, applying full QAT to VGGT-1B is extremely resource-intensive:
> * VGGT-1B [1] is trained on large-scale multi-view data (millions of images from more than 17 datasets) with heavy computation.
> * The original training of VGGT-1B requires **> 500 A100 GPU-days**, which is well beyond the typical deployment budget for PTQ scenarios.
>
> **A small-scale QAT study on QuantVGGT.**
> Due to the limited computation resources but to still provide empirical evidence, we conduct a lightweight QAT experiment by fine-tuning the quantized model for **1 epoch** on the **same calibration dataset** used by QuantVGGT (i.e., no extra data are introduced). The results are:
>
> |Method|GPU Memory (GB)|GPU Time (Hours)|AUC@30$\uparrow$|AUC@15$\uparrow$|AUC@5$\uparrow$|AUC@3$\uparrow$|
> |-|-|-|-|-|-|-|
> |Full Prec.|-|-|90.5|84.4|67.9|57.0|
> |**QuantVGGT (PTQ)**|14.6|2.67|87.9|79.0|58.4|44.7|
> |**QuantVGGT (QAT)**|53.7|9.50|88.1|80.2|58.6|44.6|
>
> We draw the following observations:
> * **QAT indeed improves low-bit performance.** Even with only 1 epoch on a small calibration subset, QAT brings consistent gains over pure PTQ (e.g., +1.2 AUC@15, +0.2 AUC@30), confirming that **applying QAT to QuantVGGT is effective and has clear potential** to further close the gap to the full-precision model.
> * **The computational overhead is substantial.** Achieving these modest gains requires **~3.7× GPU memory** (14.6 → 53.7 GB) and **>3.5× GPU time** (2.67 → 9.50 hours) even in this very limited setting, whereas our main results focus on efficient PTQ with minimal resource requirements.
>
> **Positioning relative to our work.**
> Our paper is intentionally centered on **efficient PTQ** that can be deployed without revisiting large-scale training. The above experiment shows that:
> * QAT on top of QuantVGGT is **meaningful and beneficial** for low-bit quantization;
> * It is also **computationally much more demanding**, making it a different regime than the deployment-oriented PTQ setting we target.
>
> We will add this discussion and the table to the revised manuscript Table 5, and explicitly clarify that **QAT is complementary rather than conflicting** with our method: QuantVGGT provides a strong PTQ baseline that can serve either as a standalone solution or as **a good initialization for subsequent QAT**, depending on the available computational budget.
>
> >**W4:** Visualizations and empirical sensitivity analysis for Theorem 3.2.
>
> We understand the reviewer's concern. And we clarify that we have already performed both **visualizations** and **quantitative sensitivity analysis** to support Theorem 3.2. We will highlight this more clearly in the revision.
>
> 1. **Sensitivity analysis of NFDS vs. other sampling schemes.**
> We have conducted a **quantitative comparison** between the proposed NFDS and random sampling in Section 4.3 Figure 5. We also present detailed ablation study on each proposed technique including noise-filter and frame-wise cluster. We also present the results here, all the results are conducted under five different random seeds:
>
> |Method|AUC@30|
> |-|-|
> |Random|85.4$_{\pm0.96}$|
> |Filtered|86.0$_{\pm0.43}$|
> |Clustered|86.2$_{\pm0.38}$|
> |**NFDS**|**86.9$_{\pm0.22}$**|
>
> * NFDS achieves the **highest mean AUC@30** and, importantly, the **lowest standard deviation**, indicating both better performance and **higher stability**.
> * This **aligns with Theorem 3.2**, which shows that NFDS reduces the discrepancy between calibration sample distribution and the true data distribution.
> 2. **Visualizations of sampling distributions.**
> As shown in Figure 4(b)\(c)(d) in the main paper, we **visualize the sample distributions** of different strategies. NFDS tends to select calibration samples that are more **diverse and representative**, while random sampling often exhibits outliers and harder to distangle the diversity.
>
> In summary:
>
> * Theorem 3.2 provides a **formal justification** for why representative sampling matters for calibration.
> * The visualization shows **how NFDS operationalizes this theory** in practice (by favoring “well-spread” 3D configurations).
> * The sensitivity table demonstrates that this leads to **consistent performance gains with reduced variance**.
>
> Together, these elements demonstrate that Theorem 3.2 is not merely a decorative theoretical result; it directly supports the **design and empirical effectiveness** of NFDS.

---

> ### Author Response · Authors · 2025-11-22
>
> >**Q1:** Evaluation on more diverse 3D datasets.
>
> Please see the reply on ***W1 & Q1***.
>
> >**Q2:** Perform under mixed precision or asymmetric quantization.
>
> We further evaluate QuantVGGT under mixed precision and asymmetric quantization under W4A4 on CO3Dv2. We use the first 200 sequences of each class for faster evaluation:
>
> |Method|AUC@30$\uparrow$|AUC@15$\uparrow$|AUC@5$\uparrow$|AUC@3$\uparrow$|
> |-|-|-|-|-|
> |Full Prec.|90.5|84.4|67.9|57.0|
> |QuaRot|82.8|71.4|41.0|24.4|
> |**QuantVGGT**|87.9|79.0|58.4|44.7|
> |**QuantVGGT (*Mixed Precision*)**|88.0|78.8|58.4|44.6|
> |**QuantVGGT (*Asymmetric*)**|87.8|78.9|58.6|44.8|
>
> We observe that:
> * All three variants of QuantVGGT (symmetric W4A4, mixed precision, and asymmetric) achieve **very similar performance**, and all significantly outperform QuaRot across all metrics.
> * The small performance differences between the three QuantVGGT variants suggest that our design already **effectively mitigates activation-range imbalance and outliers**, so the method is **robust to the choice of symmetric vs. asymmetric and uniform vs. mixed-precision** formats.
>
> These results indicate that QuantVGGT is not tied to a specific quantization format and can be **readily deployed under mixed-precision or asymmetric settings** while maintaining strong accuracy.
>
> >**Q3:** Sensitivity to Hadamard matrix and smoothing coefficient $\alpha$.
>
> For Hadamard matrix, as it is randomly generated, we use different random seed for ablation. And we vary smoothing $\alpha$ for sensitivity analysis. Here, we present the results:
>
> |Method|AUC@30$\uparrow$|AUC@15$\uparrow$|AUC@5$\uparrow$|AUC@3$\uparrow$|
> |-|-|-|-|-|
> |Full Prec.|89.5|83.2|66.1|54.9|
> |QuaRot|81.8|70.3|40.1|23.5|
> |**QuantVGGT (seed 1, $\alpha=0.5$)**|86.9|78.7|57.3|43.6|
> |**QuantVGGT (seed 1, $\alpha=0.3$)**|86.6|78.4|57.0|43.3|
> |**QuantVGGT (seed 1, $\alpha=0.7$)**|86.5|78.3|57.0|43.2|
> |**QuantVGGT (seed 2, $\alpha=0.5$)**|86.8|78.6|57.1|43.5|
> |**QuantVGGT (seed 3, $\alpha=0.5$)**|86.9|78.7|57.3|43.7|
>
> * Varying α in range [0.3, 0.7] causes at most only 0.4 AUC@30 fluctuation, and other metrics behave similarly. All $\alpha$ significantly **outperforms** current SOTA method QuaRot. In our experiment, we select $\alpha=0.5$ as a balanced choice.
> * Changing the Hadamard seed has almost no effect (AUC@30 varies within 0.1). This indicates the **random Hadamard matrix is stable** which proves the effectiveness of our proposed DSFQ.
>
> These results indicate that QuantVGGT hyperparameter does **not rely on heavily tuning** and is **robust to small perturbations**.
>
> We add this sensitivity table and emphasize this robustness in the revised paper Table 10.

---

> ### Author Response · Authors · 2025-11-22
>
> >**Q4:** Experiments on other 3D backbones.
>
> To validate QuantVGGT generality across architectures, we additionally evaluate QuantVGGT on **Fast3R** [1], a recently proposed transformer-based 3D reconstruction backbone. We report point-cloud reconstruction results on 7-Scenes [2] dataset:
>
> |Method|Acc. Mean$\downarrow$|Acc Med.$\downarrow$|Comp. Mean$\downarrow$|Comp. Med.$\downarrow$|N.C. Mean$\uparrow$|N.C. Med.$\uparrow$|
> |-|-|-|-|-|-|-|
> |Full Prec.|0.049|0.021|0.069|0.021|0.624|0.683|
> |SmoothQuant|0.497|0.448|0.319|0.244|0.586|0.625|
> |QuaRot|0.312|0.258|0.149|0.080|0.593|0.637|
> |**QuantVGGT**|**0.049**|**0.022**|**0.070**|**0.022**|**0.624**|**0.683**|
>
> QuantVGGT achieves **practically lossless performance** relative to the full-precision Fast3R model and **substantially surpasses** strong quantization baselines (QuaRot and SmoothQuant) on all metrics.
>
> These results highlight the following points:
> 1. **DSFQ targets a universal outlier problem, while being especially well aligned with VGGT.**
> The design of DSFQ is directly motivated by the severe heavy-tailed outliers observed in VGGT, which are particularly amplified by VGGT’s special tokens. In this sense, DSFQ is naturally tailored to VGGT. At the same time, its mechanism depends only on empirical activation statistics: it rescales and regularizes heavy-tailed outliers regardless of where the outliers arise from. The Fast3R results show that our DSFQ achieves practically lossless performance and clearly surpasses SmoothQuant and QuaRot, demonstrating that this VGGT-motivated design in fact resolves a universal outlier issue shared by many large 3D transformers.
> 2. **NFDS exploits intrinsic 3D structure, while being naturally matched to VGGT’s inductive bias.**
> NFDS is derived from our sampling principle (Theorem 3.2) and only assumes that the inputs are **multi-view 3D sequences**. It uses frame-wise correlation metrics for diverse sampling and depends solely on the natural 3D geometry and temporal structure of the data. VGGT’s special tokens and first–subsequent frame design amplify these frame-wise correlations, making NFDS particularly effective and well aligned with VGGT’s inductive bias. Importantly, when we apply QuantVGGT to Fast3R, we still observe **lossless performance**, whereas generic PTQ baselines degrade substantially. This confirms that NFDS is **not restricted to VGGT**, but benefits any 3D backbone that exhibits multi-view correlations.
> 3. **Overall message.**
> The lossless performance on Fast3R shows that our **VGGT-tailored designs remain architecture-agnostic at the principle level**:
>     * They are inspired by and particularly suitable for VGGT due to its strong inductive bias and pronounced outlier behavior;
>     * Yet they **address fundamental, model-agnostic PTQ challenges**: universal activation outliers (DSFQ) and 3D correlation-aware calibration (NFDS). And therefore can be directly transferred to other 3D backbones.
>
> [1]. Yang J, et al. Fast3r: Towards 3d reconstruction of 1000+ images in one forward pass. CVPR 2025.
>
> [2]. Shotton J, et al. Scene coordinate regression forests for camera relocalization in RGB-D images. CVPR 2013.

---

### Official Review · Reviewer_KMES · 2025-11-01

**Soundness:** 3
**Presentation:** 3
**Contribution:** 2
**Rating:** 8
**Confidence:** 3

**Summary:**

The paper tackles PTQ of billion‑parameter VGGT, a recent 3D reconstruction backbone. The authors diagnose two obstacles for VGGT PTQ: (i) heavy‑tailed activations induced by data‑independent special tokens (camera & register tokens), and (ii) unstable calibration due to the multi‑view nature of 3D inputs. They propose Dual‑Smoothed Fine‑Grained Quantization (DSFQ) and Noise‑Filtered Diverse Sampling (NFDS) to address the issues.

**Strengths:**

1. The paper provides convincing evidence that VGGT’s data‑independent registration tokens produce heavy‑tailed, high‑variance channels that break naïve PTQ; the visualizations in Fig. 3 and Fig. 7 make this concrete.
2. DSFQ is well‑motivated: Hadamard rotation preserves matmul but spreads outliers, and the subsequent channel scale is computed after rotation, avoiding pre‑scale instability. The choice of token‑wise activation is sensible for transformer matmuls and validated in Table 5.
3. Strong empirical results at low bit‑widths. W4A4 QuantVGGT clearly surpassing generic baselines.

**Weaknesses:**

1. The baseline set is strong (GPTQ, SmoothQuant, QuaRot, DopQ‑ViT), but some competitive activation‑aware baselines (e.g., AWQ‑style or ViT‑specific PTQ variants) are not included; also unclear if all baselines received equal calibration size/selection tailored to 3D sequences.
2. External validity beyond VGGT. Claims focus on VGGT‑1B and VGGT is a great model. However, it is unclear whether the observations and designs can translate to other 3D backbones.

**Questions:**

Have you tried QuantVGGT on models without VGGT’s special tokens, eg. Fast3R? Does NFDS remain effective?

---

> ### Author Response · Authors · 2025-11-22
>
> We are grateful for your positive assessment of our contributions and for your helpful suggestions. We respond to your questions as follows.
>
> >**W1:** Comparison of more quantization baselines and calibration fairness.
>
> Following your suggestion, we compare our QuantVGGT with AWQ [1] and RepQ-ViT [2] under the same W4A4 setting on CO3Dv2 datasets with 10 frames:
>
> |Method|AUC@30$\uparrow$|AUC@15$\uparrow$|AUC@5$\uparrow$|AUC@3$\uparrow$|
> |-|-|-|-|-|
> |Full Prec.|89.5|83.2|66.1|54.9|
> |AWQ|81.2|69.9|40.8|26.5|
> |RepQ-ViT|82.4|69.9|37.1|20.2|
> |**QuantVGGT**|**86.9**|**78.7**|**57.3**|**43.6**|
>
>
> These results show that **QuantVGGT outperforms both AWQ and RepQ-ViT by a large margin at all metrics**. This confirms that our VGGT-tailored design (DSFQ + NFDS) provides clear benefits over generic activation-aware or ViT-specific PTQ approaches when dealing with the heavy-tailed, multi-view statistics of large 3D reconstruction transformers.
>
> **Calibration protocol and fairness.**
>
> To address the concern on calibration fairness:
> * All PTQ baselines and QuantVGGT are calibrated with **exactly the same calibration budget**: 40 sequences sampled from the Co3Dv2 training split, following the setting described in Sec. 4.1 and Appendix B.
> * For data selection, **QuantVGGT uses NFDS** to construct a diverse and informative calibration subset, while **all other methods use random sampling over the same pool** and therefore share the same calibration data.
>
> We will clarify this unified calibration protocol and the use of NFDS versus random sampling in the revised manuscript.
>
> [1]. Lin J, et al. Awq: Activation-aware weight quantization for on-device llm compression and acceleration. MLsys 2024.
>
> [2]. Li Z, et al. Repq-vit: Scale reparameterization for post-training quantization of vision transformers. ICCV 2023.

---

> ### Author Response · Authors · 2025-11-22
>
> >**W2 & Q1:** Experiments on other 3D backbones.
>
> To validate QuantVGGT generality across architectures, we additionally evaluate QuantVGGT on **Fast3R** [1], a recently proposed transformer-based 3D reconstruction backbone. We report point-cloud reconstruction results on 7-Scenes [2] dataset:
>
> |Method|Acc. Mean$\downarrow$|Acc Med.$\downarrow$|Comp. Mean$\downarrow$|Comp. Med.$\downarrow$|N.C. Mean$\uparrow$|N.C. Med.$\uparrow$|
> |-|-|-|-|-|-|-|
> |Full Prec.|0.049|0.021|0.069|0.021|0.624|0.683|
> |SmoothQuant|0.497|0.448|0.319|0.244|0.586|0.625|
> |QuaRot|0.312|0.258|0.149|0.080|0.593|0.637|
> |**QuantVGGT**|**0.049**|**0.022**|**0.070**|**0.022**|**0.624**|**0.683**|
>
> QuantVGGT achieves **practically lossless performance** relative to the full-precision Fast3R model and **substantially surpasses** strong quantization baselines (QuaRot and SmoothQuant) on all metrics.
>
> These results highlight the following points:
> 1. **DSFQ targets a universal outlier problem, while being especially well aligned with VGGT.**
> The design of DSFQ is directly motivated by the severe heavy-tailed outliers observed in VGGT, which are particularly amplified by VGGT’s special tokens. In this sense, DSFQ is naturally tailored to VGGT. At the same time, its mechanism depends only on empirical activation statistics: it rescales and regularizes heavy-tailed outliers regardless of where the outliers arise from. The Fast3R results show that our DSFQ achieves practically lossless performance and clearly surpasses SmoothQuant and QuaRot, demonstrating that this VGGT-motivated design in fact resolves a universal outlier issue shared by many large 3D transformers.
> 2. **NFDS exploits intrinsic 3D structure, while being naturally matched to VGGT’s inductive bias.**
> NFDS is derived from our sampling principle (Theorem 3.2) and only assumes that the inputs are **multi-view 3D sequences**. It uses frame-wise correlation metrics for diverse sampling and depends solely on the natural 3D geometry and temporal structure of the data. VGGT’s special tokens and first–subsequent frame design amplify these frame-wise correlations, making NFDS particularly effective and well aligned with VGGT’s inductive bias. Importantly, when we apply QuantVGGT to Fast3R, we still observe **lossless performance**, whereas generic PTQ baselines degrade substantially. This confirms that NFDS is **not restricted to VGGT**, but benefits any 3D backbone that exhibits multi-view correlations.
> 3. **Overall message.**
> The lossless performance on Fast3R shows that our **VGGT-tailored designs remain architecture-agnostic at the principle level**:
>     * They are inspired by and particularly suitable for VGGT due to its strong inductive bias and pronounced outlier behavior;
>     * Yet they **address fundamental, model-agnostic PTQ challenges**: universal activation outliers (DSFQ) and 3D correlation-aware calibration (NFDS). And therefore can be directly transferred to other 3D backbones.
>
> [1]. Yang J, et al. Fast3r: Towards 3d reconstruction of 1000+ images in one forward pass. CVPR 2025.
>
> [2]. Shotton J, et al. Scene coordinate regression forests for camera relocalization in RGB-D images. CVPR 2013.

---

### Author Response · Authors · 2025-11-24
**General Response**

Dear AC and Reviewers,

We sincerely thank all reviewers for their time and effort in evaluating our paper. We are pleased that the reviewers acknowledged the following contributions:

* **Strong motivation.** We are, to the best of our knowledge, the first to systematically analyze VGGT-specific PTQ challenges, including heavy-tailed activations from camera/register tokens and instability under multi-view 3D inputs. *[Reviewers KMES, 48HL, o5ar, 7NMK, and 9UAK]*
* **Tailored methodology.** Our DSFQ and NFDS modules are explicitly designed for VGGT’s characteristics, directly addressing outlier smoothing and calibration sampling rather than relying on generic PTQ tricks. *[Reviewers KMES, 48HL, o5ar, 7NMK, and 9UAK]*
* **Theoretical grounding.** The rotation/smoothing formulation and NFDS sampling principle are supported by formal analysis that is closely linked to the algorithm design and observed behavior. *[Reviewers KMES, o5ar, 7NMK, and 9UAK]*
* **Comprehensive experiments.** We provide extensive comparisons to strong PTQ baselines and detailed ablations across datasets, tasks, bit-widths, and backbones, demonstrating the robustness of QuantVGGT. *[Reviewers KMES, o5ar, 7NMK, and 9UAK]*
* **Practical efficiency.** QuantVGGT achieves near full-precision quality at low bit-widths while delivering substantial memory and latency savings, facilitating deployment of large 3D transformers in real-world systems. *[Reviewers 48HL, o5ar, 7NMK, and 9UAK]*
* **Clarity and visualization.** The paper’s structure, activation/token visualizations, and overall presentation were found to clearly convey the main ideas and insights. *[Reviewers o5ar, 7NMK, and 9UAK]*

We also thank the reviewers for their constructive suggestions, which led to the following major revisions:

* Added broader activation-aware and ViT-specific PTQ baselines and clarified unified calibration settings for all methods. *[Reviewer KMES]*
* Evaluated QuantVGGT on additional datasets, tasks, and backbones (including more 3D tasks and non-VGGT models) to strengthen generalization claims. *[Reviewers KMES, 48HL, o5ar, and 7NMK]*
* Reported more real-hardware cost details. *[Reviewers 48HL, o5ar, 7NMK, and 9UAK]*
* Conducted calibration-budget and stability ablations, including low-sample regimes and more comparisons between NFDS and random sampling. *[Reviewer 48HL, o5ar, and 7NMK]*
* Included a lightweight QAT baseline and discussed its trade-offs relative to our PTQ-focused design. *[Reviewer 48HL, o5ar, and 7NMK]*
* Refined theoretical exposition, notation, and visualizations, and expanded background on Hadamard rotation and SmoothQuant to improve readability. *[Reviewers 48HL, o5ar,  7NMK and 9UAK]*

All these changes have been incorporated into the revised manuscript and are highlighted in red. We also provide detailed point-by-point responses to each reviewer.

Best regards,

Authors of Paper #2786

---

### Meta-Review · Area_Chair_op8w · 2026-01-07

**Summary:**

Quantized Visual Geometry Grounded Transformer (QuantVGGT) targets post-training quantization for billion-scale VGGT 3D reconstruction models. It diagnoses two VGGT-specific PTQ failures: heavy-tailed activations driven by data-independent camera/register tokens, and unstable calibration when inputs are multi-view sequences. It proposes DSFQ (Hadamard pre-rotation plus post channel smoothing with fine-grained quantization) and NFDS (deep-layer outlier filtering plus frame-aware diverse clustering) to stabilize ranges and calibration. Results show large gains over generic PTQ baselines at W4A4, near-lossless quality, and real-hardware speed and memory wins. I recommend accept.

**Reviewer Concerns:**

Addressed: missing activation-aware / ViT PTQ baselines and calibration fairness (added AWQ/RepQ-ViT and unified calibration), generalization beyond VGGT (added Fast3R transfer plus extra 3D datasets like 7-Scenes/NRGBD), real-hardware details, low-calibration-size and stability ablations (including 5-sample regime), QAT reference baseline, and sensitivity to Hadamard/smoothing and other hyperparameters.

Remaining: limited breadth of deployment evaluation (one GPU setup, limited inference stacks) and still incomplete comparison to other acceleration families.

**Reviewer Scores:**

KMES: likely stays at 8 (accept) after baseline + fairness clarification and Fast3R transfer evidence. 7NMK: likely stays at 8 given added W4A8, extra datasets, and stronger evidence for token effects and first-frame bias. 48HL and o5ar: likely move from 6 to 7 because hardware clarification, calibration-size ablations, QAT reference, and broader evaluation reduce key doubts. 9UAK: likely moves slightly up (6 to 7) after fixes, sequence-length latency, and added visual comparisons.

---

### Decision · Program_Chairs · 2026-01-26

Accept (Poster)